# Neutron and Gamma-Ray Detection System Coupled to a Multirotor for Screening of Shipping Container Cargo

**DOI:** 10.3390/s23010329

**Published:** 2022-12-28

**Authors:** Luís Marques, Luís Félix, Gonçalo Cruz, Vasco Coelho, João Caetano, Alberto Vale, Carlos Cruz, Luís Alves, Pedro Vaz

**Affiliations:** 1Centro de Investigação da Academia da Força Aérea, Academia da Força Aérea, Instituto Universitário Militar, Granja do Marquês, 2715-021 Pêro Pinheiro, Portugal; 2Instituto de Plasmas e Fusão Nuclear, Instituto Superior Técnico, Universidade de Lisboa, Av. Rovisco Pais 1, 1049-001 Lisboa, Portugal; 3Centro de Ciências e Tecnologias Nucleares, Instituto Superior Técnico, Universidade de Lisboa, Estrada Nacional 10 (km 139.7), 2695-066 Lisboa, Portugal

**Keywords:** unmanned aerial vehicle, security and defense, neutron, beta- and gamma-ray detection, plastic scintillators, silicon photomultipliers, mobile radiation detection system

## Abstract

In order to detect special nuclear materials and other radioactive materials in Security and Defense scenarios, normally, a combination of neutron and gamma-ray detection systems is used. In particular, to avoid illicit traffic of special nuclear materials and radioactive sources/materials, radiation portal monitors are placed at seaports to inspect shipping-container cargo. Despite their large volume (high efficiency), these detection systems are expensive, and therefore only a fraction of these containers are inspected. In this work, a novel mobile radiation detection system is presented, based on an EJ-200 plastic scintillator for the detection of gamma rays and beta particles, and a neutron detector EJ-426HD plastic scintillator (with ^6^Li) embedded in a compact and modular moderator. The use of silicon photomultipliers in both detectors presented advantages such as lightweight, compactness, and low power consumption. The developed detection system was integrated in a highly maneuverable multirotor. Monte Carlo simulations were validated by laboratory measurements and field tests were performed using real gamma-ray and neutron sources. The detection and localization within one meter was achieved using a maximum likelihood estimation algorithm for ^137^Cs sources (4 MBq), as well as the detection of ^241^Am–beryllium (1.45 GBq) source placed inside the shipping container.

## 1. Introduction

### 1.1. Radiological and Nuclear Threats

Illicit traffic of special nuclear materials (SNMs) and radioactive sources and materials is a cause for concern worldwide, due to the possible use of these materials in improvised nuclear devices (INDs) and radiological dispersal devices (RDDs) or radiological exposure devices (REDs). Large and heavy fixed radiation portal monitors (RPMs) are normally used at international borders, sea ports and airports in order to detect SNMs and radioactive sources or materials. RPMs are normally used to screen shipping-container cargo and vehicles. Portable RPMs can also be deployed for security screening (e.g., major events) and contamination monitoring, e.g., population monitoring after a radiological or nuclear (RN) incident [1]; however their mobility is reduced, and they are also heavy.

Despite the fact that many radionuclides are used as radioactive and radiation sources in industry, medicine and research, only a few of them are widely available in concentrated amounts that could be used in RDDs, namely: ^241^Am, ^252^Cf, ^137^Cs, ^60^Co, ^192^Ir, ^238^Pu, ^210^Po, ^226^Ra, and ^90^Sr [2]. Three of these radionuclides, ^137^Cs, ^60^Co and ^192^Ir, emit gamma rays with energies in the hundreds of keV or slightly above 1 MeV (for ^60^Co), whilst ^241^Am is considered a low-energy gamma-ray emitter (59.5 keV energy line). ^90^Sr is a beta emitter, and like the alpha emitters ^241^Am, ^252^Cf, ^238^Pu, ^210^Po, and ^226^Ra, is dangerous mainly when ingested or inhaled. Neutron sources, such as ^252^Cf (spontaneous fission source) or ^241^Am–beryllium (which results from the mixing of an alpha emitter with a light nucleus such as beryllium), are used for soil and concrete density and moisture measurements, and in the oil and gas well logging industry.

For Security and Defense, SNMs are a major concern, since the detonation of an IND would cause not only the dispersal of radioactive material, but would also lead to mass casualties. While plutonium can be detected by both gamma rays and neutrons (spontaneous fission source), highly enriched uranium (HEU) is extremely difficult to detect, since: (i) low-intensity and low-energy gamma rays (185 keV) are emitted, which can be easily shielded, concealed or masked; and (ii) it is characterized by a very low emission rate of neutrons [3]. Active interrogation (using X-rays, gamma rays, neutrons or muons) is an alternative to passive detection equipment when sources are not detectable (weak or shielded sources) [4,5].

### 1.2. Screening of Shipping Containers

Since the sea freight corresponds to around 90% of traded goods worldwide, there is a challenge related to the screening of shipping-container cargo at seaports due to the volume and speed of trade flows. Therefore, maintaining the normal flow of legitimate goods and at the same time, undertaking the monitoring of nuclear and radioactive sources, as well as other illegal imports (e.g., explosives, narcotics and conventional weapons), can be a very difficult and challenging task given the number of front-line officers (FLO) available [6].

To facilitate transportation via ships, rails or trucks, standard steel containers are used, which are 20 foot and 40 foot long. Some inspection techniques are available based on the cargo documentation check, a physical search of the container (which is very time consuming), and by deploying non-intrusive imaging (e.g., X-rays or gamma rays). Despite the advantages of non-intrusive imaging, such as faster detection times and preselection of containers for physical searches, expensive equipment is used and operation and maintenance costs have to be taken into account. For example, in European ports, only about 10% of incoming containers are scanned, and of these, only 2% are physically searched [7]. While a complete scan of a container can take, on average, less than one minute, in the case of an alarm, the secondary inspection is normally carried out manually with a handheld radioisotope identification device (RIID) that may take up to twenty minutes. If the secondary inspection is inconclusive, a third and more exhaustive inspection is made by certified radiation experts and implies unpacking the container (this may take 3 h for a 40-foot long container). Scanning transshipments is also a challenging task, since the containers are offloaded from one ship and loaded into another ship without passing through the RPMs (which are normally located at the seaport exit/entrance points) [6].

According to Martin and Connolly [8] a well-designed screening system should efficiently detect and identify radioactive materials and SNMs that could be used to fabricate a RDD or an IND, while keeping the normal flow of cargo affordable so that it can be easily replicated. At the seaports, this is normally achieved via a two-stage process: (i) a primary inspection in which the shipping containers pass through a large polyvinyl-toluene (PVT) plastic scintillators (high geometric detection efficiency) for gamma-ray detection and, in some cases, neutron detectors based normally in ^3^He tube detectors. A “counts above threshold” alarm criteria is used to select the containers that will be further inspected; (ii) a secondary inspection for source identification and localization. In order to make better decisions, the following capabilities are highlighted as being of paramount importance: the use of data processing algorithms such as the energy windowing, rapid radionuclide identification, activity estimation and source localization. Some alternatives to fixed RPMs are a network of mobile or stationary high-resolution inorganic scintillators, for use in urban and border monitoring scenarios, that could be used not only for screening purposes but also for source characterization and localization. At seaports, this network of detectors could be transported by port vehicles and continuously map the radiation.

Algorithms described in literature for the detection and, in some cases, localization of radioactive sources using a detector network are: the maximum likelihood estimation (MLE) [9,10], triangulation radiation source detection (TriRSD), sequential probability ratio test (SPRT), source-attractor radiation detection (SRD) [10] and the particle filter (PF) [11,12]. In [9] a MLE algorithm could estimate the localization of a 189 kBq ^137^Cs source with an accuracy of 0.53 m from the measurements of five ϕ5.1 cm × 5.1 cm NaI(Tl) detectors (fixed network) considering a 5 × 5 × 5 m^3^ parameter space and an interval of 3 min. For a source of approximately 22.6–34 MBq, the authors predict a 1 s measurement time for the source localization. In [12], the use of a PF algorithm allowed the authors to estimate the 281 kBq ^137^Cs source localization with accuracy of 1.5 m (in a 10 × 10 m^2^ area) using the available experimental data from the intelligent radiation sensor systems (IRSS) tests of 22 stationary ϕ5.1 cm × 5.1 cm NaI(Tl) scintillators (measurement time of 5 min).

A shipping container screening system must feature the highest possible true positive rate (TPR, related to the detection sensitivity), and at the same time, the lowest false-positive rate (FPR). To reduce the FPR, it is necessary to distinguish the radiological and nuclear threats from naturally occurring radioactive material (NORM) (^40^K, uranium and thorium decay series present in some materials), medical isotopes or at a smaller scale background variation.

Another cause for concern is the procurement of the chemical elements, compounds of mixtures used by the deployed radiation detection system and technology. Due to the worldwide shortage of the isotope ^3^He, it is necessary to find an alternative technology for the neutron-detection systems. Some security-related requirements for the replacing technology are: (i) high neutron detection efficiency; (ii) ability to detect both fast and slow neutrons, as neutrons might be attenuated by some shielding or by the cargo materials; and (iii) the lowest possible gamma-ray sensitivity (to avoid false alarms) [13].

### 1.3. Mobile Radiation Detection Systems

As previously mentioned, in security scenarios, we are interested in detecting gamma rays, beta and alpha particles and neutrons. However, due to their long range in air, the detection of gamma rays and neutrons is preferable for mobile radiation detection systems. Since the range in air of the beta particles of ^90^Sr and ^90^Y are in the order of some meters (maximum beta range in air of ^90^Y is approx. 10.6 m) [14], a mobile beta particle detection system should also be considered in platforms that operate near to the ground, such as cars, multirotors and handheld equipment.

According to [15], in illicit traffic of nuclear and other radioactive materials scenario a combination of gamma-ray and neutron detection systems are normally used. Due to the large stand-off distances, possible weak and/or shielded radioactive sources, large detection systems (∼1 m^2^) are used. In order to transport these radiation detection systems and contextual sensors, mobile platforms such as cars, vans and trucks are used. Examples of projects which developed a combination of radiation detection systems include the radiological multi-sensor analysis platform (Rad_Map) [16,17], the sistema mobile per analisi non distruttive e radiometriche (SLIMPORT) [18], the modular detection system for special nuclear material (MODES_SNM) [19], the mobile urban radiation search (MURS) [20], and the real-time wide area radiation surveillance system (REWARD) [21,22,23,24]. In MODES_SNM, the detection system was also tested in the scanning of maritime containers as a primary control device (next to RPMs), as a secondary control (inspection of containers that already triggered an alarm in a RPM) and by using radioactive samples for identification purposes at Rotterdam seaport. Difficulties related to natural background variation were reported. The system was able to detect and identify gamma-ray sources and NORM, as well as neutron sources such as ^252^Cf, ^241^Am–beryllium (hereafter designated Am-Be), Pu-Be, SNM (Pu and U samples) and the presence of hydrogenated or lead shielding [19]. The use of a dual-mode Cs_2_LiYCl_6_ (CLYC) scintillator with ^6^Li (simultaneous detection of gamma-rays and thermal neutrons by using pulse shape discrimination techniques) allows compact and lightweight detection systems which can be coupled with a multirotor [25,26,27,28]; however, the sensitive volume has only 12.86 cm^3^. Table 1 summarizes the main characteristics of mobile radiation detection systems used in security scenarios (in some cases also applied to radiation safety scenarios) such as area monitoring, mapping and source localization and identification in urban environments.

A network of low-cost mobile detectors with georeferenced data was also proposed for source localization in an urban environment [29].

**Table 1 sensors-23-00329-t001:** Combination of mobile radiation detection systems used in security scenarios.

Gamma-Ray Detection System	Fast/Thermal Neutron Detection System	Mobile Platform	Project/Ref.
NaI(Tl) imager; HPGe	EJ-309 liquid scintillators (fast)	Truck	Rad_Map [16,17]
NaI(Tl) ^1^; LaBr3(Ce)	NE-213 liquid scintillator (fast); ^3^He proportional counter (thermal)	Not specified	SLIMPORT [18]
Xe scintillator	^4^He scintillator (fast); ^6^Li-lined ^4^He tubes (fast and thermal)	Van	MODES_SNM [19]
NaI(Tl)	^6^LiF (thermal)	Car	MURS [20]
-	EJ-309 liquid scintillator (fast); BF_3_ and ^3^He detectors with HDPE (thermal)	Truck	[30]
Two stacked 1 cm^3^ CdZnTe (CZT)	Thin planar silicon PIN diodes covered with hydrogenated plastic radiators (fast); Silicon backfilled with ^10^B (thermal)	Not specified	REWARD [21,22,23,24]
CZT (1 cm^3^); CLYC with ^6^Li ^2^	CLYC with ^6^Li (thermal)	Multirotor	[27,28]
CLYC with ^6^Li ^2^	CLYC with ^6^Li (thermal)	Multirotor	[25,26]

^1^ Large NaI(Tl) detector used to detect energetic gamma rays (6 MeV) originated by active techniques (tagged neutrons). ^2^ Dual-mode detector.

The choice of the right mobile platform for the radiation detection system will have an impact on the radiation measurements’ quality and effectiveness. Some requirements that should be considered when selecting the mobile platform are the weather sensitivity, payload capacity, cost, ease of operation, and spatial resolution obtained during radiation mapping measurements. The mobile platform can be ground based (terrestrial or maritime), air based or hybrid, and each of them can be manned or unmanned (teleoperated, semi-autonomous or autonomous operation). The use of unmanned platforms allows us to avoid unnecessary radiation exposure risks to humans, perform autonomous mapping and monitoring, and is a more cost-beneficial solution than manned platforms [15].

The ground-based platform solutions have the advantage of greater payload capacity and autonomy; however, obstacles on the ground can limit their operation and normally require greater data-collecting times compared to air-based platforms [15].

The use of an unmanned aerial vehicle (UAV), such as a multirotor (also known as a drone), to carry a small unmanned ground vehicle (UGV), could help to overcome the obstacles on the ground (e.g., emergency scenarios) and then the UGV could perform the survey in greater detail [31].

The literature also refers to the combined use of an UAV and an UGV (cooperative operation). To improve the path planning and hotspot localization of the UGV, the UAV could provide photogrammetry (3D terrain reconstruction) and a broader area radiation mapping [32]. In [33], the use of an UGV to improve the navigation accuracy of an UAV is also described.

Unlike manned aircraft, UAVs allow operation at lower altitudes and speeds, improving the spatial accuracy in radiation measurements. The advantage of using multiple low-cost UAVs (e.g., cooperative radioactive search or a swarm of UAVs) over a single-UAV approach was also demonstrated for low-altitude source localization and contour mapping, in particular for urgent radiation detection (e.g., emergency scenario) and for large areas [34,35,36]. Challenges in security and safety scenarios (e.g., nuclear accident mitigation) such as all-terrain and confined spaces operation (e.g., mountain or urban areas) and the search for low-activity sources can be overcome with the use of multirotor platforms [35]. Multirotors are easy to operate, are very maneuverable, and have vertical take-off and landing (VTOL) and hovering capabilities; however, their payload is limited to a few kilograms [15].

Recent literature refers to the use of compact gamma-ray detection systems coupled with multirotors, normally to obtain radiation mapping of contaminated areas, such as areas near to the Fukushima Daiichi Nuclear Power Station (FDNPS) or legacy uranium mines, as well as to detect, localize and identify radioactive sources (Table 2). CZT is the most commonly used detector, but its small volume limits its use for weak or shielded gamma-ray sources (small solid angle). A good alternative to CZT is, for example, the inorganic scintillator CsI(Tl) with SiPM readout; however, a commercial SIGMA50 detector is limited to 32.8 cm^3^ and the energy resolution is 7.2% at 662 keV [37,38]. The use of a SiPM-based scintillator was also demonstrated for the detection of radioactive sources in scrap metal (waste and recycle material monitoring) when strong magnetic fields (0.1 T) are present [39]. Due to its high density and atomic number, BGO detectors are very sensitive to gamma rays; however, they feature poor energy resolution and are very heavy (a total sensitive volume of 206 cm^3^ weights 4 kg).

Multirotors are also referred in the literature as a platform suitable for carrying lightweight Compton cameras (gamma-ray imaging); however, they are used in radiation safety scenarios, and the measurements were obtained inside radioactive contaminated buildings of the FDNPS (with significant radiation intensity) [56,57].

In [15] the use of multiple UAVs was suggested, firstly to detect and localize a radioactive source(s) using plastic detectors (poor energy resolution but low price) and afterwards, in a second phase, to use an inorganic scintillator for identification purpose. Moreover, plastic scintillators are lighter materials and can be manufactured in several shapes so that they can be used in small platforms with payload restrictions such as multirotors. In Table 3, some advantages and limitations of plastic scintillators are highlighted.

The use of high-Z sensitized plastic scintillators using organometallics or nanocomposites is an active research area. Considering the use of organometallics, the addition of bismuth to the plastic scintillator formulation improves its spectroscopy capability but degrades the light yield. The use of iridium complex fluors improve the light yield of plastic scintillators for counting purposes [58].

In order to convert the scintillation light produced by the interaction of gamma rays and charged particles (primary or secondary) within the detection sensitive volume into electrical signals, silicon photomultipliers (SiPMs) are quickly replacing the photomultipliers tubes (PMTs) technology, in particular for mobile applications [15]. Unlike PMTs, SiPMs are very compact, lightweight, require low bias voltage (normally 5 V), low power consumption and are immune to magnetic fields interference.

In this work, a novel radiation detection system is proposed, consisting of:A larger cross sectional area EJ-200 plastic scintillator for gamma-ray and beta particles detection (improving solid angle in measurements performed at a distance, instead of the heavy, smaller and more expensive semiconductors and inorganic scintillation crystals);A plastic scintillator EJ-426HD with ^6^Li content. A compact and modular high-density polyethylene (HDPE) moderator for neutron detection (thermal and fast component) was added;SiPM readout for both plastic scintillators;A highly maneuverable multirotor platform used to carry the radiation detection system. This platform allows hovering, VTOL and offers the ability to fly at very low altitudes and speeds. By reducing the source–detector distance, an increase in the overall geometric detection efficiency is also obtained;Able to simultaneously detect gamma rays, beta particles and neutrons, as well as to perform source characterization and localization.

The mobile radiation detection system is composed of a ϕ110 mm × 30 mm EJ-200 plastic scintillator (285 cm^3^) and a EJ-426HD (with ^6^Li content) plastic scintillator with a modular HDPE moderator sheets, both with SiPM readout, that were developed and tested by the authors for screening shipping-container cargo.

The working principle of a neutron detection system based on the ^6^Li isotope is related to the following thermal neutron capture reaction (cross section of 940 barns) [59]:(1)$${}^{6}Li+n\to^{4}He{+}^{3}H+4.78MeV$$

To the best of the authors’ knowledge, no work has been carried out using plastic scintillators with SiPM readout coupled with a multirotor for the screening of shipping-container cargo.

The proposed mobile radiation detection system can be used to detect and localize SNMs and other radionuclides inside shipping containers, acting as the primary inspection device (as an alternative or complement to RPMs) or as secondary inspection device when the container triggers an alarm at RPMs and is subject to a more exhaustive search (currently performed by handheld equipment). For a fast detection on the primary inspection phase a lateral wall screening of the container (drone at half the height) will be performed and tested with a time to inspect lower than 50 s. If more time is available for the inspection, for example in a secondary inspection, the following characteristics will be assessed: (i) the benefits of lateral wall screening of the container at different heights or; (ii) a complete turn to the container will also provide information about the source localization inside the container using a maximum likelihood estimation (MLE) algorithm. It must be highlighted that the secondary inspection performed by the developed mobile radiation detection system (few minutes) allows a significant time reduction compared to the inspection performed by handheld equipment, and avoids unnecessary exposure risks to humans.

Compared to other mobile radiation detection systems, this solution presents advantages such as lower costs, compactness, light weight (and consequently, more flight time available), an increase in overall detection efficiency due to the significant increase in the geometric detection efficiency and source–detector distance reduction (using a multirotor).

This detection system can also be used to screen other infrastructures (e.g., urban environments) or vehicles. If the detection system is reoriented 90°, it is possible to map contaminated areas and search for lost sources on the ground. The first results showed that the mobile radiation detection system can detect and localize a 4 MBq ^137^Cs source within one meter, and can detect a mixed source with 1.45 GBq Am-Be and 215 MBq ^137^Cs (shielded or not) placed inside a shipping container.

## 2. Materials and Methods

### 2.1. Mobile Radiation Detection System Development

The developed prototype of a radiation detection system comprises two independent plastic scintillators. A third detector, an inorganic scintillator CsI(Tl), was also used in laboratory tests to allow the comparison of results. All detectors were manufactured by Scionix (including the SiPMs integration on the scintillators) [60] and the specifications are resumed as follows:Gamma-ray and beta particles detector—cylindrical in shape with a 110 mm diameter and 30 mm thick EJ-200 plastic scintillator [61], with a built-in bias generator/preamplifier and four 12 × 12 mm^2^ SiPMs (arrays J-60035-4P-PCB). To improve beta particles’ detection sensitivity, a 32 micron titanium entrance window was added. It has an additional power connector in its housing to feed the neutron detector. Weight: 517 g (short cable included);Neutron detector—with parallelipedic shape, consisting of two layers of 25 × 90 mm^2^ and 0.32 mm thick EJ-426HD (with ^6^Li content) and a wavelength shifter EJ-280 (25 × 90 × 4 mm^3^) [61] placed between them with a built-in bias generator/preamplifier and three 6 × 6 mm^2^ SiPMs (KETEK PM6660). Two connections are available: (i) for both detector signal and SiPM power; and (ii) a TTL counting output—each TTL pulse corresponds to a neutron count (detector internally adjusted above noise at 40 °C). Weight: 95 g (short cables included);Gamma-ray detector (only used in laboratory tests)—51 mm diameter and 51 mm thick CsI(Tl) scintillator with a built-in temperature compensated bias generator and preamplifier, two 12 × 12 mm^2^ SiPMs (arrays J-60035-4P-PCB) and an aluminum housing. Weight: 600 g (short cable included).

The detectors’ size and arrangement were chosen according to three aims: to maximize the detection efficiency, not exceed the platform’s maximum take-off weight and fit on the carbon fiber sandwich sheet developed to carry the gamma and neutron detection system side-by-side.

Figure 1 illustrates a scheme of the connections between the detectors and associated electronics. TOPAZ-SiPM multichannel analyzer (MCA), developed by BrightSpec [62], with power consumption of approximately 1.1 W has three input connectors: (i) a Lemo connector (type ERN.03.302.CLL) to read the detector analog signals and to provide the necessary power to the SiPMs integrated on the scintillators (5V, 20 mA); (ii) a Lemo connector (type ERN.00.250.CTL) for programmable general purpose input/output (GPIO) signals (can be used as an external counter input); and (iii) a USB type mini B for data output, device power supply and control using, for example, a Raspberry Pi model 3B. TOPAZ-SiPM MCA combines, in a small and lightweight box (70 mm × 45 mm × 26 mm, 70 g), the following features: analog-to-digital converter (ADC) with a spectral memory size up to 4096 channels, analog signal amplification (up to 16), a traditional trapezoidal shaper for digital pulse processing, a digital baseline restorer and a pile-up rejector and a 5V low-ripple (low-noise) power supply for the SiPMs preamplifiers. Since only one TOPAZ-SiPM MCA was available, to simultaneously read the gamma-ray/beta and neutron detection system signals, it was necessary to connect the EJ-200 scintillator to the analog input of TOPAZ-SiPM MCA (Lemo connector type ERN.03.302.CLL) and the EJ-426HD neutron detector (TTL output) to the GPIO input of TOPAZ-SiPM MCA (Lemo connector ERN.00.250.CTL). When using the TTL output of the EJ-426HD neutron detector, its analog output (LEMO connection) is only used for power-supply purposes (connected to a +5 V power plug available in the EJ-200 housing). In order to obtain the energy spectrum of the EJ-426HD neutron detector, it is also possible to connect its analog output into TOPAZ-SiPM MCA (only used on laboratory tests); however, in this case, EJ-200 cannot be connected to TOPAZ-SiPM MCA (analog connector already in use).

In order to obtain a standalone radiation detection system which could be easily integrated into any mobile platform or used as handheld equipment, we chose to use an independent power supply (power bank) and a global navigation satellite system (GNSS) antenna. The radiation detection system and associated electronics have a power consumption of approximately 2.75 W (550 mA current, 5 V). Using a power bank of 10 Ah, a battery life of up to 18 h was obtained.

To improve the position accuracy of the radiation measurements, more expensive alternatives to a single GNSS antenna could be explored in the future, such as a real-time kinematic (RTK) GNSS or a differential GNSS.

Some advantages could also be found relating to the use of the electronic components that might be available on the mobile platform (alternative hardware architecture), such as: (i) radiation measurement data transmission to a ground control station or receiving navigation instructions using telemetry antennas; (ii) position accuracy improvement and redundancy using the telemetry data provided by the GNSS antenna(s) and the inertial measurement unit (IMU); and (iii) payload weight reduction and increase in the platform’s autonomy, in particular for UAVs, by using the platform’s power supply (normally batteries) and GNSS antenna(s). However, this architecture is platform dependent and requires a broader comprehension of the platform hardware (e.g., a power-supply adapter) and software (e.g., communication protocols, telemetry data access and integration on the radiation detection system).

The Raspberry Pi model 3B was remotely accessed via Wi-Fi using a laptop and a dedicated router for hardware initialization, start/stop data acquisition and to access the stored radiation measurement data, which were timestamped and georeferenced.

For the EJ-200 scintillator, sampling times of 1 s, 2 s and 4 s were used for the handheld configuration, while for the detection system integrated in the drone, a sampling time of 1 s was used. In order to optimize the statistics of the neutron detection system measurements (results presented only for the drone configuration), larger integration times were chosen, which corresponded to the time spent carrying out the shipping container’s screening process: approximately 50 s for lateral wall screening and 120–140 s for a complete turn screening.

Since the EJ-426HD detector is mostly sensitive to thermal neutrons, it was necessary to develop a compact and lightweight moderator in order to detect fast neutrons. A moderator made of four parallelepipedic sheets of HDPE (20 mm thickness each) was developed (Figure 2) and optimized for an Am-Be source, using Monte Carlo (MC) modeling and simulations, as well as experimental tests [63]. The EJ-426HD detector is embedded in the central moderator sheets (1 cm thickness is used to accommodate the detector), resulting in a moderator total thickness of 7 cm and a cross-sectional area of 14.5 × 11 cm^2^. Since the moderator is modular, it is possible to increase the detection efficiency for moderated sources (e.g., shielded by hydrogenous materials) by: (i) removing the peripheral sheets (reducing payload weight); or (ii) by changing the position of the peripheral sheets; for example, moving one moderator sheet (located between the potential source and detector) to the opposite side, i.e., increasing the reflector thickness (with no payload weight change).

In order to integrate the EJ-200 and EJ-426HD detectors in mobile platforms, a carbon fiber sandwich sheet with 20 × 30 cm^2^ was manufactured. Supports for both detectors were 3D printed using polylactic acid (PLA) filament (Figure 3).

A handheld configuration was developed for easy data acquisition and comparison with the data obtained by the drone (Figure 4).

The developed mobile radiation detection system, with a total weight of 2.8 kg (associated electronics, supports and the carbon fiber sandwich sheet included), was also integrated in a DJI Matrice 600 Pro (Figure 5), a hexacopter with a maximum take-off weight of 15.5 kg (considering a 6 kg maximum payload) [64].

### 2.2. Software Architecture

The software developed can be divided via the following steps:1.Clocks synchronization. Before running the radiation data-acquisition code, it is necessary to synchronize the Raspberry Pi clock using the GNSS receiver clock [65];2.GNSS data acquisition. The National Marine Electronics Association (NMEA) GGA messages are read each second (GNSS antenna receiver) and the timestamped information is stored (the timestamp is converted in unix time) in an output file (“GNSS information”);3.Radiation data acquisition. Using the simulation development kit libraries provided by the TOPAZ-SiPM MCA manufacturer, a code is developed in order to provide:Hardware initialization: several parameters have to be loaded to TOPAZ-SiPM MCA in order to read the detector’s signals, such as course gain, fine gain, number of channels and MCA model by selecting pulse-height analysis (PHA) or multi-channel scaling (MCS), time of acquisition (for PHA mode) or dwell time (for MCS mode), upper- and lower-level discriminator, rise time, flat top, digital base line restorer, and pile-up rejector. PHA mode is used for the laboratory tests, while for the field tests, MCS mode is selected;Start the radiation data acquisition;Data acquisition and storage. A timestamp using the Raspberry Pi clock (in unix time) for each radiation measurement is stored in the Raspberry Pi memory card. Because the EJ-200 plastic scintillator and EJ-426HD neutron detection system have different integration times, two output files (“Radiation data”) are created.Stop data acquisition (if no predefined acquisition time has been inserted);4.Data processing. This step is carried out after data acquisition and consists of the following steps:The first step consists of searching for the same timestamp values in the “GNSS information” file and in the “radiation data” file and merging the desirable information, which is the latitude and longitude (in degrees), altitude (in meters), and radiation intensity (in cps). A comma-separated values (CSV) file is produced by running a simple python code. When the integration time (dwell time) of the radiation measurement is higher than 1 s, it is considered the middle value of the GNSS timestamp for position purposes. For example, for 2 s and 4 s dwell times (gamma-ray measurements) the considered GNSS timestamp is the one corresponding to 1 s and 2 s after the measurement initiates, respectively;For radioactive source localization, the Matlab program is used to read the CSV file and convert the latitude and longitude (in degrees) to universal transverse mercator (UTM) coordinates. In order to simplify the graphics, all “x” and “y” UTM coordinates are subtracted by their minimum “x” and “y” value, respectively, and therefore are presented in relative units. After that, all radiation intensity data points are processed by a MLE algorithm for a single source position estimation [42,66]. Finally, the distance between the true source position and the MLE estimated source position is calculated;For radioactive source detection, the radiation intensity points stored in the CSV file are plotted against the time of occurrence and compared to a decision threshold [67] given by backg_mean + 1.645 × SD, where backg_mean corresponds to the average of the background radiation points measured around the container (with no source), and SD is the standard deviation of these points.

The MLE algorithm used in this work uses a likelihood function based on a Poisson distribution of the propagation model of radiation. The estimated position is the position where the likelihood function has the highest value. The likelihood function is derived as a function of the position and is equal to zero. For better computational efficiency, the maximum likelihood is calculated using the logarithm. This approach only considers one radioactive source [42,66].

The data processing stage is performed offline; however, in future works, the algorithm will work online for near real-time decisions while the mobile platform is navigating. Advantages such as early warning (radioactive source detection and localization) and informative path planning (adapt predefined paths according to the measurements) could be achieved.

Figure 6 summarizes the software architecture.

### 2.3. Monte Carlo Simulation

The state-of-the-art MC simulation program MCNP6 [68] was used to compare the gamma-ray detection efficiency of the EJ-200 plastic scintillator with a commercial CsI(Tl) scintillator considering different source-to-detector distances (geometric detection efficiency) and the intrinsic detection efficiency of each detector.

For the MC simulations, the following parameters were used:Detectors’ dimensions and material: (i) ϕ110 mm × 30 mm EJ-200 plastic scintillator (density 1.023 g/cm^3^); (ii) ϕ51 mm × 51 mm CsI(Tl) scintillator (density 4.51 g/cm^3^);Tally F8 (pulse height distribution);Energy thresholds: Due to the different light yield of the detectors, an energy threshold of 11 keV for CsI(Tl) and 55 keV for EJ-200 (five times higher) were considered. The energy threshold for CsI(Tl) was obtained experimentally via energy calibration;Sources considered separately: ^137^Cs with gamma-ray energy of 662 keV and ^241^Am with gamma-ray energy of 59.54 keV. Point sources were isotropic and centered at the detector window;Physics: model e p (tracking of electrons and photons during particle transport simulation);Intermediary medium: air.

Since the MCNP6 F8 tally type gives the gamma rays detected per starting particle, to obtain the detection efficiency, one must multiply the F8 tally value by the radiation yield. For the 662 keV gamma rays of ^137^Cs the yield is 0.8499 (i.e., approximately 85 gamma rays of 662 keV are emitted per 100 disintegrations) while for the 59.5 keV gamma rays of ^241^Am, the yield is 0.3592 [69,70].

The total detection efficiency depends on both the intrinsic and geometric efficiency. Due to the higher density and atomic numbers of the detection volume, inorganic scintillators such as CsI(Tl) have higher intrinsic efficiency than the plastic scintillators. However, when the source to detector distance increases, the higher cross-sectional area of EJ-200 contributes to a significant increase in the geometric detection efficiency and, consequently, on the total detection efficiency (as shown in Figure 7). Three source–detector distances were considered in the MC simulations: 1 mm (source attached to the detector window), 1 m and 5 m.

## 3. Results

### 3.1. Radiological Measurements

The developed radiation detection system was tested in three phases:Laboratory tests;Field tests using the “handheld configuration” of the radiation detection system (setup of Figure 4);Field tests using the radiation detection system integrated into the drone (setup of Figure 5).

The laboratory tests consisted of placing low-intensity sources of 22 kBq (0.60 µCi) ^241^Am, 8.5 kBq (0.23 µCi) ^137^Cs, and 3.3 kBq (0.09 µCi) ^90^Sr next to the entrance window of the EJ-200 plastic scintillator (see Figure 8a) and CsI(Tl) scintillator and obtaining the corresponding spectra for a specific integration time. Since the EJ-200 detector has a diameter higher than its thickness, the variation of the gamma-ray counts (for a given integration time) was obtained, varying the angle of the ^137^Cs source position and keeping the source to detector center distance constant equal to 30 cm (as indicated in Figure 8b). For the neutron detector, a 33 MBq (0.9 mCi) Am-Be source was placed at 2 cm distance of the detector (as shown in Figure 8c) and a spectrum was obtained.

For the field tests, radioactive sources were positioned at the center or at the corner of an empty standard 20-foot long shipping container. Two types of radioactive sources were used. The first one consisted of ten equal sources of ^137^Cs with a total activity of 4 MBq (0.11 mCi) (see Figure 9a). The second one consisted of Troxler equipment (chosen due to difficulties in the procurement of a neutron source with the desirable activity) [71] oriented 90° (see Figure 9b) in which a 1.45 GBq (39.2 mCi) Am-Be source (with an uncertainty of ±10%) and a collimated 215 MBq (5.81 mCi) ^137^Cs source (with an uncertainty of ±10%) can be found. Two configurations were considered when using the Troxler: (i) safe position—the ^137^Cs is shielded by lead and tungsten; (ii) first notch after the safe position—the ^137^Cs becomes unshielded in the bottom side of the equipment (the tungsten sliding block moves to the side). In both configurations, the Am-Be source is emitting neutrons.

The field tests consisted of moving the radiation detection system along the lateral walls of the shipping container (as shown in Figure 10) at approximately constant speed and at half the container height (1.3 m). For the detection system integrated in the drone, we also considered movements at one-third of the container height (0.864 m ≈ 0.86 m) and two-thirds of the container height (1.73 m). The radiation detection system also performed complete turns around the shipping container for radioactive source detection improvement and source localization purposes. For each radiation detection system screening the height and position of the source, the experiment was repeated five times for data validation.

When using the handheld configuration, the motion of the radiation detection system was performed with an approximate speed of 0.33 m/s and at a constant distance of 1 m relative to the shipping container lateral walls. With the detection system integrated in the drone, the speed was reduced to approximately 0.2 m/s.

Since the drone was being operated manually (for safety reasons, mainly related to the proximity to the shipping walls and wind conditions), it was not possible to maintain a constant distance between the drone and the shipping walls. To account for the detection system to container distance variations and improve the source detection and localization accuracy, light detection and ranging (LiDAR) equipment could be used. The LiDAR would allow us to obtain: (i) the detection system to container wall distance; (ii) the detection system to ground distance; and (iii) the detector’s orientation relative to the container. Alternately, the LiDAR could be used to keep the detection system to container wall distance constant by increasing the drone navigation accuracy along the path. Despite the many advantages of a LiDAR, some drawbacks, such as the increase in the payload weight and the higher power consumption, must be also considered.

Because all radiation data points are timestamped and georeferenced, it is possible to see a more irregular path when using the drone.

Figure 11 shows a snapshot of the screening of a 20-foot long shipping container performed by the radiation detection system integrated in the multirotor.

For the considered radiation detection system (payload), a flight time of 17–22 min was achieved (depending on the path performed by the drone and battery pack used).

### 3.2. Monte Carlo Simulations

Table 4 displays the MC simulation results of the F8 tally values obtained for the EJ-200 and CsI(Tl) detectors considering point sources of ^241^Am and ^137^Cs (simulation parameters described in Section 2.3).

Due to the energy threshold of 55 keV, the EJ-200 detector detection efficiency for the 59.5 keV gamma rays of ^241^Am is very small compared to the weight equivalent CsI(Tl) detector.

When considering the situation of a ^137^Cs source attached to the detector window (1 mm distance), the CsI(Tl) detector features a higher detection efficiency; however, when the source is at 1 m and 5 m distance, the EJ-200’s detection efficiency is 1.59 and 1.47 times higher than the CsI(Tl), respectively (due to a higher geometric detection efficiency).

### 3.3. Laboratory Tests

#### 3.3.1. Neutron Detection System

In Figure 12 a typical spectrum obtained with the EJ-426HD using a 33.3 MBq (0.9 mCi) Am-Be source is shown. A neutron rate of 14 counts/s was obtained. By surrounding the Am-Be source with a lead cylinder (to absorb the gamma-rays of ^241^Am) no significant change in the count rate was observed, as expected.

#### 3.3.2. Gamma Ray and Beta Particle Detection System

In this subsection a comparison between the detection efficiency (gamma-rays and beta particles) of the developed EJ-200 plastic scintillator and a CsI(Tl) scintillator is presented.

Figure 13 shows the spectrum of both detectors considering a ^90^Sr source (beta emitter) attached to the detector window. Considering the total counts with the background subtracted, the EJ-200 scintillator measured 56,253 ± 259 counts, while the CsI(Tl) scintillator measured 33,463 ± 262 counts. This means that the EJ-200 scintillator features a beta detection efficiency 1.68 times higher than the CsI(Tl) scintillator.

Since ^241^Am has peaks at low gamma-ray energy (59.5 keV and a lower intensity peak at 26.3 keV), the spectra for both detectors were obtained in order to analyze their lower energy thresholds (see Figure 14). From Figure 14a, it is clear that EJ-200 scintillator cannot detect the gamma rays of ^241^Am, while CsI(Tl) can detect both the 59.5 keV and 26.3 keV peaks of ^241^Am. EJ-200 has an energy threshold of approximately 55 keV, which is very close to the 59.5 keV gamma-ray peak of ^241^Am; therefore, the small fraction of gamma rays that hit the detector and deposit energy above 55 keV is residual and can be easily masked by the background variation.

Figure 15 shows the spectra obtained with EJ-200 and CsI(Tl) scintillators for a ^137^Cs source next to the detector window. Due to the lower intrinsic efficiency of EJ-200 scintillator (lower atomic number and density), it features a detection efficiency that is a factor of 0.60 smaller than the CsI(Tl) scintillator. This result is compatible with the value 0.62 ± 0.01 obtained by Monte Carlo simulations (refer to Table 4). The CsI(Tl) detector showed an energy resolution of 6.8% at 662 keV.

According to the experimental setup of Figure 8b, a gamma-ray counts variation with the source–detector angle was obtained (shown in Figure 16). Due to the presence of the neutron detector and moderator material between the source and the EJ-200 scintillator, a reduction of almost 10% and 25% in the detection efficiency was observed at the angles 160° and 180° compared to the symmetric angles 20° and 0°, respectively.

### 3.4. Field Tests

Despite the fact that EJ-200 plastic scintillator and the EJ-426HD neutron detection system performed simultaneous measurements, in order to allow for a comprehensive presentation of the results, this section was divided as follows:Gamma-ray and neutron background measurements;Neutron detection system measurements;Gamma-ray detection system measurements.

#### 3.4.1. Gamma-Ray and Neutron Background Measurements

Figure 17 represents the gamma-ray count rate of the background (sampling time of 1 s) obtained using the handheld configuration surrounding the shipping container. The trajectory of the radiation detection system is represented with relative units. An average gamma-ray background of 17.1 cps and a standard deviation of 4.8 cps was obtained.

To determine the neutron background count rate, several measurements were performed around the shipping container. Each measurement had integration times of 80 s on average. This resulted in a neutron background mean of 0.07 ± 0.03 cps (Figure 18).

#### 3.4.2. Neutron Detection System Measurements

Table 5 and Table 6 display the average and the standard deviation of the five measurements performed by the neutron detection system coupled with the drone when:Screening the shipping container laterally (one side only; the same side as the Troxler equipment bottom face);Performing a complete turn around the shipping container.

Considering the Troxler equipment at the center of the container (Table 5), the neutron count rate is higher when the drone’s motion is performed at half of the container’s height.

By positioning the Troxler equipment at the bottom corner of the shipping container (Table 6), a higher neutron count rate is obtained when the drone’s motion is performed at one-third of the container’s height.

The higher neutron count rate values obtained at half of the container height for Table 5 and one-third of the container height for Table 6 match the approximate neutron source position (source placed at the center and bottom corner of the container, respectively). Therefore, a shipping container inspection performed at three different heights can provide an approximate neutron source position estimate.

Notice that the maximum neutron count rate values found in both situations (0.27 ± 0.06 cps and 0.45 ± 0.13 cps for lateral side screening) are well above the neutron background (0.07 ± 0.03 cps), which creates a high likelihood of the presence of a neutron source.

#### 3.4.3. Gamma-Ray Detection System Measurements

Using the handheld configuration of the radiation detection system, the gamma-ray count rate considering different sampling times (dwell times) was obtained with ^137^Cs sources of 4 MBq centered and at the bottom corner of the shipping container (examples in Figure 19 and Figure 20, respectively). Applying a MLE algorithm, it was possible to observe that for sampling times of 2 s and 4 s, due to the increase in statistics (gamma-ray counts in each point), the estimation of the sources position improves when considering centrally placed sources.

Table 7 summarizes the average localization error and SD for all measurements performed. For the sources placed at the center of the container, the distance D between the MLE estimated position and the true source position is less than one meter, while for the sources placed at the corner, the distance D can reach 1.3 m.

Despite the fact that the localization estimate improves with the increase in the sampling time, when considering sources that are not centrally positioned, the consequent loss of points might represent less measurement data for the algorithm to better estimate the source position—as can be seen from the small amount of improvement of the distance D with the sampling time when the sources are placed at the corner. Therefore, to obtain a similar effect (and not lose data points), we decided to use a sampling time of 1 s and reduce the speed survey from approximately 0.33 m/s (speed used to perform the handheld configuration measurements) to 0.2 m/s for the detection system coupled with the drone.

The results of the gamma-ray detection system coupled with the multirotor are displayed in Figure 21, Figure 22, Figure 23, Figure 24, Figure 25, Figure 26, Figure 27, Figure 28 and Figure 29.

Since only the lateral side of a shipping container is available for screening purposes (this happens most of the time), we decided to screen a lateral side with the radiation detection system coupled with the drone at different heights, along a pathlength of about 10 m (6 m of the container lateral side plus 2 m for each side) keeping the drone as close as possible from the container walls. Since the drone was operated manually, distances relative to the container wall from 1 m up to 3 m were observed due to manual adjustments and wind conditions.

Figure 21 and Figure 22 shows the screening performed on the container lateral wall at half the container height when the ^137^Cs sources of 4 MBq are in the center and at the bottom corner of the shipping container, respectively. The x axis is related to the time elapsed since the beginning of the container screening. Different colors were used to distinguish between the five screenings performed.

Despite the significant source–detector distance (up to 4 m), the iron shielding of the container walls and the sampling time of 1 s, it was possible to detect the 4 MBq ^137^Cs sources when placed at the center of most of the measurements (Figure 21), i.e., almost all measurement dataset points were above the decision threshold (background mean + 1.645 × SD). Since for the heights 0.86 m and 1.3 m, not all measurements triggered an alarm, it can be said that this situation corresponds to the minimum detectable activity (MDA). This MDA depends not only on the source activity and the source–detector distances considered, but also the drone speed and sampling time used.

The ^137^Cs of 4 MBq placed at the closest bottom corner of the container (corner located in the container wall side where the screening is taken place) could be easily detected (Figure 22), i.e., all measurement datasets have points significant higher than the decision threshold. The peak registered on the graphs of Figure 22 at approximately 15 s (in particular, the screening performed at one-third of the container height) corresponds to the time the drone is closer to the source (passing through the source).

In order to determine the localization capability of this gamma-ray detection system, five complete turns to the shipping container for each height considered were performed with the ^137^Cs sources of 4 MBq placed at the center and at the bottom corner of the shipping container. Some examples of source localization estimation using the MLE algorithm are shown in Figure 23 and Figure 24, and the average of the distance values between the estimated and the true source position are summarized in Table 8.

While Figure 23 shows a good agreement between the estimated position and the true position of the ^137^Cs sources centered in the shipping container (distance lower than 1 m), when considering the sources placed at the corner, distances up to 2.5 m between the estimated and the true sources position can be observed (Figure 24). This can be partially explained by the lack of radiation measurement points with significant intensity (near to the sources) in an asymmetric radiation distribution, and by the position error of the detection system (visible in Figure 24 for the height 1.73 m, where the measurement points are on top of the true source position).

When considering the Am-Be source and the shielded 215 MBq ^137^Cs source (Troxler equipment oriented 90° in safe position), gamma-ray detection using the lateral measurements is possible (Figure 25) but like the situation of the ^137^Cs sources of 4 MBq centered at the container, it corresponds to approximately the MDA of the detection system. However, according to Section 3.4.2, the neutron count rate detected at the lateral wall (0.27 cps) is well above the neutron background (0.07 cps). Therefore, it is of paramount importance to combine a gamma-ray and neutron detection system in order to confidently trigger the alarm.

Moreover, if container cargo material was considered around the radiation source, more gamma-ray and neutron attenuation would take place (depending on the material density and atomic number) reducing the counting rate measured outside and, consequently, the MDA and the radiation source localization accuracy.

In order to gain source characterization and localization, the gamma-ray count rate for complete turns to the container at half its height was obtained (examples can be found in Figure 26). Figure 26 shows an example of source localization where it is possible to observe some leaks of radiation from the Troxler equipment, which gives higher count rate values at the container lateral wall which is behind the top of the equipment. This lateral wall corresponds to the opposite side analyzed in Figure 25. The distances (D) average between the estimated source position and true source position was 1.32 ± 0.47 m.

Considering the Am-Be source and the collimated 215 MBq ^137^Cs source unshielded only the radiation measurements performed during complete turns at different heights were analyzed. Due to the high gamma-ray intensity of this source it is easily detected, even performing a path with the drone some meters far from the lateral wall where the collimated radiation is incident. In Figure 27 it is shown some examples of localization estimation. Since the ^137^Cs source is highly collimated by the tungsten and lead shielding (on the lateral and top side of the equipment respectively), an asymmetric radiation distribution arises and the MLE algorithm give misleading information about the Troxler position. The distance between the true Troxler position and the MLE estimated position can achieve almost 3 m (as shown in Table 9), which is higher than the smaller lateral dimension of the container (2.44 m).

A 3D view of the shipping container screening can be obtained by adding the georeferenced radiation data of the complete turns acquired at different heights for each source location. Figure 28 shows the 3D view of the 4 MBq ^137^Cs sources at the center and at the bottom corner of the shipping container and in Figure 29 a 215 MBq ^137^Cs collimated source. For better visualization of the count rate observed at different heights, the data aspect ratio for the z axes was reduced compared to that of the x and y axes. From Figure 28 and Figure 29, it is possible to estimate the approximate location of the radioactive source by considering, for instance, the higher gamma-ray count rate in a given path location.

Since the gamma-ray emission of the 4 MBq ^137^Cs sources are, in principle, isotropic, the slightly higher gamma-ray count rate in one lateral compared to the opposite side at a height of 1.3 m, when the sources are in the center (left Figure 28), can be justified by: (i) a closer drone trajectory to the lateral wall in one side; (ii) lower speed of the drone when passing in some regions of the lateral wall (it was not always possible keep the speed constant); (iii) the gamma-ray count rate measured is of the same order as the background, and therefore it can be easily influenced by statistical fluctuations.

When considering the ^137^Cs sources of 4 MBq at the bottom corner of the container, a rough estimation of the sources position due to the higher gamma-ray count rate shown at the corner of the path at 0.83 m height can be inferred.

From Figure 29, it is possible to infer the presence of a collimated gamma-ray source and the direction of the gamma rays. This information is very useful for the radiation experts when further analyses are necessary (e.g., opening the container for inspection) or to establish a safe area around the container. Due to the higher count rate on the path at a height of 1.3 m, it is also possible to infer its approximate location (in the center of the container).

## 4. Discussion

In order to study and assess the detection response and gamma-ray sensitivity of the EJ-200 plastic scintillator, a comparison with weight-equivalent CsI(Tl) scintillator MC simulations and modeling was performed using the state-of-the-art computer program MCNP6, as well as laboratory tests. Some important findings are summarized and discussed as follows. For the laboratory tests:Using an Am-Be source (activity 33.3 MBq) close to the EJ-426HD neutron detector, no significant change in the count rate was registered due to the ^241^Am gamma rays. This very low gamma-ray sensitivity of EJ-426HD is an important requirement for security applications. Despite the fact that the neutron moderator is optimized for Am-Be sources, it is modular, and the peripheral HDPE sheets can be removed (reducing weight) or the position of the peripheral sheets can be changed (e.g., placing the peripheral sheets next to each other in the opposite side of the radiation source; this allows us to reduce the moderation thickness and increase the reflector thickness);Due to the high energy threshold (approx. 55 keV) of the EJ-200 scintillator, it was not possible to distinguish the gamma rays of a 22 kBq ^241^Am source (next to the detector window) from the background. The choice of this energy threshold is related to the low light yield of plastic scintillators and the need to avoid dark counts in SiPM (which is temperature dependent). This is in accordance with MC simulations in which the detection efficiency of the EJ-200 scintillator proved to be only a small fraction (0.0035) of the detection efficiency of CsI(Tl) scintillator;For a 8.5 kBq ^137^Cs source next to the detector window, the EJ-200 scintillator showed 0.60 times smaller efficiency than the CsI(Tl) scintillator, value that is consistent with the results obtained by MC simulations. This is related to the higher density and atomic number of the CsI(Tl) detector material (which translates into a higher intrinsic detection efficiency). However, according to MC simulations, EJ-200 detectors have a factor of 1.59 and 1.47 higher detection efficiency than CsI(Tl) for source–detector distances of 1 m and 5 m, respectively. This is related to the higher geometric detection efficiency of EJ-200;When a 3.3 kBq ^90^Sr beta source was placed next to the detectors window, the EJ-200 scintillator featured a detection efficiency 1.68 times higher than CsI(Tl) scintillator.

The mobile radiation detection system was also tested in a real security scenario, for the screening of a 20-foot-long shipping container with gamma-ray sources and mixed sources (gamma-ray and neutron) inside. Two different configurations of the radiation detection system were used: handheld equipment and a drone configuration (integrated in a multirotor DJI Matrice 600 Pro).

To test the mobile radiation detection system’s ability to detect radioactive sources, a set of screenings of a shipping container lateral wall was performed. The source was detected according to a decision threshold based on the background mean and its standard deviation. For the localization of radioactive sources, complete turns to the shipping container were performed, and for each turn, a MLE algorithm was applied to the measurements’ data.

A summary of the field tests is as follows:The neutron source of Am-Be (activity 1.45 GBq) was clearly detected in all measurements by the EJ-426HD detector with 7 cm of HDPE moderator when integration times of approximately 50 s (lateral wall screening) were used. Considering the higher neutron count rate value obtained from the screening (lateral wall or complete turns) performed at three container heights, a rough neutron source localization (top, center or bottom of the container) was achieved. When considering the Troxler equipment with the ^137^Cs source shielded (safe position), the neutron detector gives a count rate of 0.27 cps ± 0.06 well above the background (0.07 ± 0.03), which gives high confidence about the detection of a potential RN threat (neutron detector complements the gamma-ray detector);For the gamma-ray source detection, a decision threshold was used to compare the data with the mean and SD of the background. The gamma-ray sources of ^137^Cs with 4 MBq placed in the container center were detected by most of the measurements performed on the lateral wall (within approximately 50 s); however the gamma-ray counting rates due to the source were close to the background, which means MDA was probably achieved. When the sources were placed at the corner of the container, all measurements allowed their detection. Despite the fact that the shielded ^137^Cs source inside the Troxler equipment was also detected in all screenings performed on the lateral wall of the container, in a real situation with more cargo inside the container, it would probably would be very difficult to detect the gamma rays. Therefore, the most efficient way to detect the Troxler equipment (mixed source) with the ^137^Cs source shielded would consist of using the neutron detection system as described before;Localization within one meter of ^137^Cs sources of 4 MBq placed at the center of the container was made possible by taking measurements with the handheld and drone configuration all around the container (120–140 s) and by using a MLE algorithm. Considering that the runtime of the MLE algorithm is only a few seconds, the localization of a 4 MBq gamma-ray source with the mobile radiation detection system would take only a few minutes (2–3 min) after the start of the data acquisition. This solution would allow us to reduce the secondary inspection times of shipping containers (which can take up to 20 min using handheld equipment), avoiding unnecessary exposure risks to humans by performing the screening with the drone configuration (remotely) and allowing for the programming of the drone to perform the screening automatically. After source localization, it is possible to place a second detector with spectroscopy capability near to that position to identify the radioisotope. When the ^137^Cs sources of 4 MBq were placed in the bottom corner of the container, the maximum distance between the MLE algorithm estimation and the true source position was 1.3 m and 2.5 m for the handheld and the drone configuration, respectively. The higher value found for the drone configuration can be explained by the greater source–detector distances performed during the screenings, which provided lower radiation intensity points. In order to achieve a better source position estimate, more data points with significant radiation intensity would be necessary; for example obtaining more data points outside the rectangle paths and at the same time, near locations where the radiation intensity is higher. This would improve the algorithm performance (giving an indication of the radiation source attenuation along a given direction). However, since this source is already in the operational limits of detection, the data points would have to be obtained not too far from the source. Therefore a trade-off arises between a better position estimate and the detection of lower activities. A localization within 1.3 m was obtained for the shielded 215 MBq ^137^Cs source (Troxler equipment in the safe position). When considering the unshielded and collimated 215 MBq ^137^Cs source, the difference between the localization estimation and the true source position can be as high as 3 m. Considering that the smaller lateral dimension of the container is 2.44 m, this distance difference means that the MLE algorithm used in this study is not adequate to deal with collimated sources, and so, other assumptions must be made. The MLE algorithm considered in this study considers an isotropic radiation emission.

The main goal of these tests was to establish proof of concept for this technology, in particular the radiation detection system coupled to a multirotor, to perform the primary and secondary inspections of shipping-container cargo. The use of a low-cost, lightweight, compact and low-power-consumption radiation detection system coupled with a highly maneuverable platform (multirotor) reduces the costs and time required for inspections (secondary inspections). The obtained results pave the way for the development of detection systems that are able to be programmed and aiming at autonomous screening operations.

## 5. Conclusions

In order to avoid the illicit traffic of SNMs and radioactive sources and materials, large RPMs are used in seaports for the screening of shipping-container cargo (using gamma ray and, in some cases, neutron detection) as well in airports and land borders. However RPMs are very expensive equipment with high operation and maintenance costs, and only a small fraction of these shipping containers are in fact scanned. The containers that trigger an alarm in RPMs (primary inspection) are then subject to time-consuming handheld inspections, which can take up to 20 min.

A network of fixed or mobile radiation detectors has been proposed; however, the existence of weak or shielded sources and the large standoff distances implies the use of several heavy detectors. Therefore mobile radiation detection systems normally used in security applications are mounted on cars or trucks. The exception is the use of dual-mode detectors such as the CLYC coupled to multirotors; however, these detectors are limited to small volumes (12.86 cm^3^).

Despite the advantages of the use of multirotors, such as high maneuverability (e.g., operate in confined spaces), VTOL and hovering capabilities, these platforms have payload-related limitations and are normally used to carry small gamma-ray detectors such as Geiger–Müller counters, semiconductors (the most common CZT) and inorganic scintillators with SiPM readout, e.g., CsI(Tl). However, Geiger–Müller counters are limited by their low gamma-ray efficiency. Meanwhile, CZT-based detection systems are very expensive and the crystal volume is typically limited to 1 cm^3^. Radiation detection systems based on CsI(Tl) technologies are also expensive, and the commercially available crystals are also limited (SIGMA50 detector has 32.8 cm^3^).

In this work, an alternative radiation detection system for the screening of shipping container cargo is presented, with the following advantages: a standalone radiation detection system, which can be easily integrated in a manned or unmanned (teleoperated, semi-autonomous or autonomous) mobile platform or used as handheld equipment; low cost; low power consumption; compact; lightweight; durability and with a high geometric detection efficiency. The solution proposed can be used for: (i) the primary inspections in which rapid detection of radioactive sources is necessary (by screening a container lateral wall). Since some containers are placed a few meters above the ground, the use of a multirotor would allow container screenings at different heights. (ii) Secondary inspections by performing a 360° screening to the lateral walls and door of the container, allowing detection of lower activities, source characterization and localization. The use of a mobile radiation detection system coupled with a drone can be an alternative or a complement to primary and secondary inspections currently performed with RPMs and handheld equipment, respectively. It can reduce the secondary inspection time, avoids unnecessary exposure risks to humans and allows autonomous inspections.

The developed mobile radiation detection system is composed of a 110 mm diameter and 30 mm thick (285 cm^3^) EJ-200 plastic scintillator with a thin titanium window for gamma-ray and beta particles detection and an EJ-426HD plastic scintillator embedded in a modular and compact HDPE moderator for the detection of fast and thermal neutrons. Both detectors have SiPM readout and are compatible with TOPAZ-SiPM MCA. This radiation detection system has an independent power supply and GNSS antenna, weights 2.8 kg and can be used as handheld equipment or integrated in a multirotor. The drone speed of 0.2 m/s was optimized to detect low-activity gamma-ray sources.

From the MC simulations and laboratory tests, the EJ-200 plastic scintillator showed advantages compared to an equivalent weight detector 51 mm diameter and 51 mm thick CsI(Tl) scintillator, such as: higher beta particle detection efficiency and higher detection efficiency to ^137^Cs gamma-rays when considering source–detector distances of 1 m and 5 m. However due to the low light yield of plastic scintillators and the use of SiPM (temperature-dependent noise), the energy threshold is normally high (above 50 keV) and the detection of ^241^Am is difficult or impractical.

Considering the laboratory and field results, the EJ-426HD neutron detector shows high detection efficiency of Am-Be neutron sources and presents a very low sensitivity to gamma-rays (which fulfills security requirements). The developed HDPE moderator is compact and modular, i.e., the peripheral HDPE sheets can be removed or can change position in order to optimize the detection efficiency for a given neutron source (moderated or not).

The developed mobile radiation detection system was tested in the screening of shipping container cargo as a proof of concept, being able to perform primary inspection within 50 s and secondary inspections in about 120–140 s (for a lateral wall screening along a 10 m path and complete turn screening, respectively—drone speed of approx. 0.2 m/s). The detection of ^137^Cs sources of few MBq and an ^241^Am-Be source (1.45 GBq) is an important fact considering the high maneuverability of the multirotor (source–detector distance reduction) and the high geometric detection efficiency of the detectors. Localization within one meter of ^137^Cs sources of 4 MBq placed at the center of the shipping container was achieved. A rough localization of the Am-Be source was also possible, inferring from the neutron count rate obtained from the screening performed at different heights. The neutron detector proved to be a good complement to the gamma-ray detector to obtain high confidence measurements of mixed radiation sources (e.g., gamma-ray and neutron emitter with the gamma-rays shielded).

The mobile radiation detection system described in this study can also be used for the inspection of other infrastructures (e.g., nuclear facilities) or vehicles. Changing the orientation of the EJ-200 detector by 90°, the mapping of a contaminated area or the search of a radioactive source on the surface can also be performed.

Some limitations of this study that need to be further addressed in future studies are related to some effects that influence the radiation measurements, such as: background variation, radiation shielding due to different cargo materials, and masking or concealing radioactive sources from, inter alia, NORM, medical isotopes’ applications, and industrial applications.

Future work shall encompass the screening of a shipping-container cargo with a ^252^Cf source (neutron spectrum is similar to that of plutonium) and to study the shielding effect of cargo-filled containers considering different materials. The use of a LiDAR should also be considered to improve the accuracy of position localization and minimize possible fluctuations. Since the main goal of the EJ-200 and EJ-426HD detection systems is fast detection and localization of SNMs and radioactive sources and materials, algorithms such as energy windowing or the use of a second payload with a CsI(Tl) scintillator (or some other high-energy-resolution detector) could be envisaged for radioisotope identification purposes. Finally, research is needed in the improvement of the light yield of plastic scintillators (e.g., new high-Z sensitized plastic scintillators) and the search for low noise SiPM (with lower temperature dependence) is desirable.

## Figures and Tables

**Figure 1 sensors-23-00329-f001:**
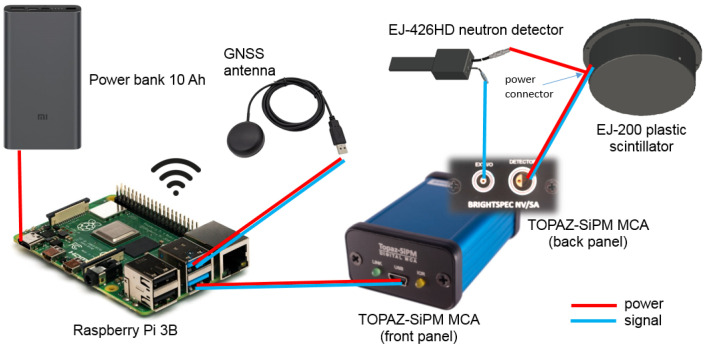
Hardware architecture. Equipment sizes are not at the same scale.

**Figure 2 sensors-23-00329-f002:**
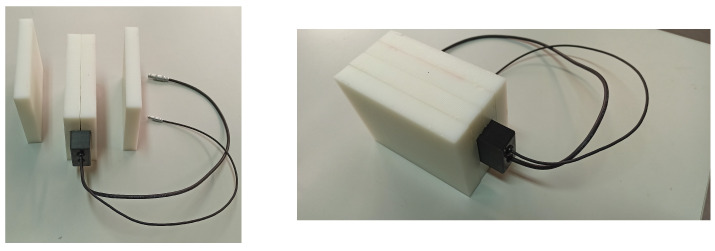
EJ-426HD detector inserted in a modular HDPE moderator with 7 cm thickness (20 mm thickness sheets of 14.5 × 11 cm^2^ cross-sectional area). Total weight: 1.2 kg.

**Figure 3 sensors-23-00329-f003:**
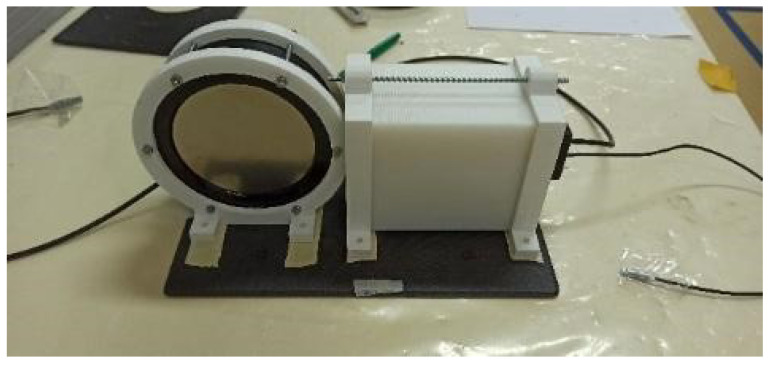
3D printed detector supports and carbon fiber sandwich sheet developed for detector integration on a mobile platform.

**Figure 4 sensors-23-00329-f004:**
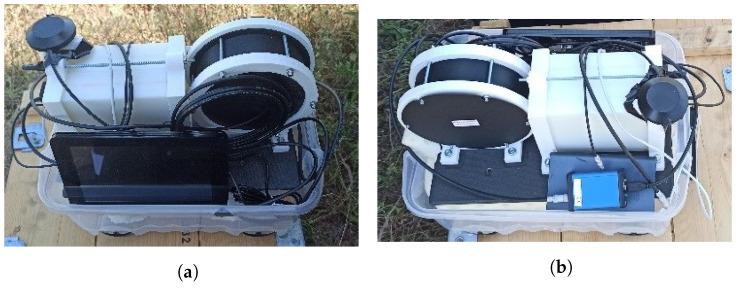
Handheld configuration of the radiation detection system: (**a**) Back side. (**b**) Front side.

**Figure 5 sensors-23-00329-f005:**
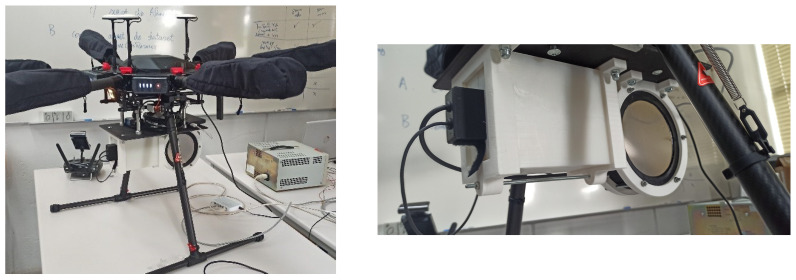
Radiation detection system integrated in the DJI Matrice 600 Pro multirotor.

**Figure 6 sensors-23-00329-f006:**
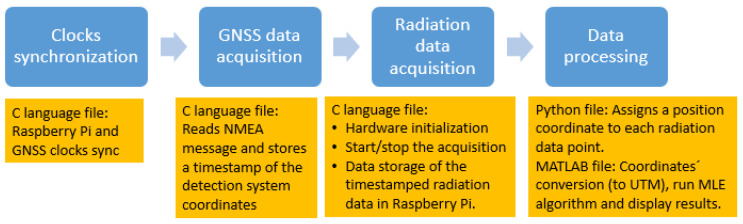
Software architecture.

**Figure 7 sensors-23-00329-f007:**
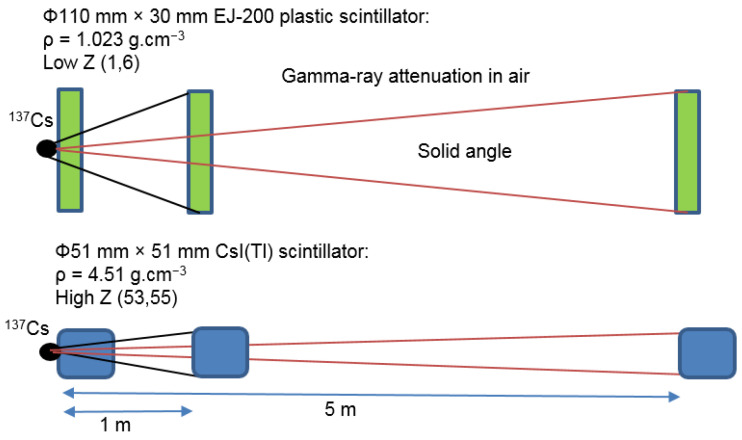
Parameters that influence the detection efficiency of EJ-200 and CsI(Tl) scintillators when considering source attached to the detector window, 1 m and 5 m distance.

**Figure 8 sensors-23-00329-f008:**
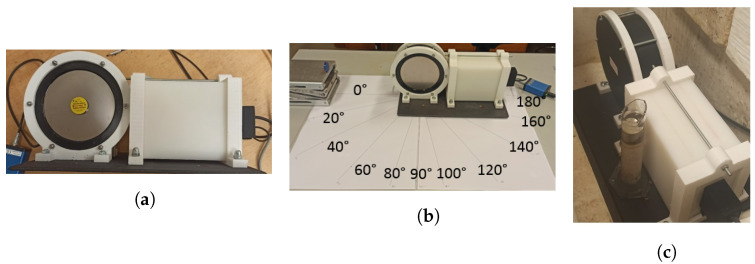
Laboratory tests performed to the developed radiation detection system. (**a**) ^137^Cs source placed next to the EJ-200 detector window. (**b**) EJ-200 response (counts) to a ^137^Cs source by varying the source–detector angle. (**c**) EJ-426HD neutron detection system spectrum acquisition using a 33 MBq (0.9 mCi) Am-Be source.

**Figure 9 sensors-23-00329-f009:**
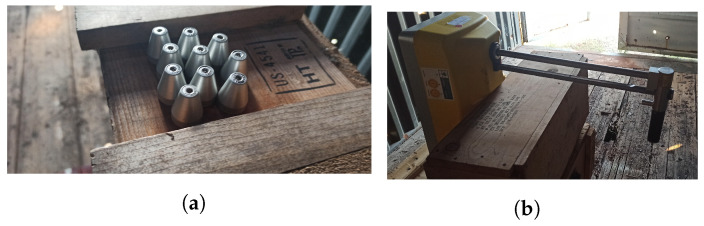
Radioactive sources used in the field tests: (**a**) ^137^Cs sources centered inside the shipping container. (**b**) Troxler equipment oriented 90° centered inside the shipping container.

**Figure 10 sensors-23-00329-f010:**
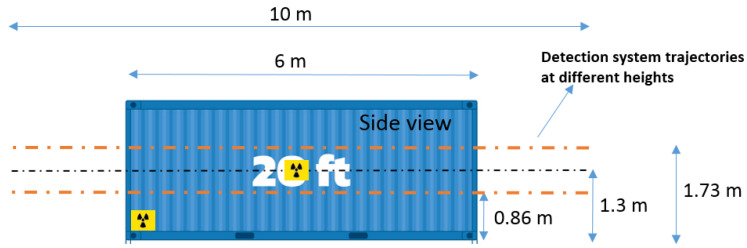
Mobile radiation detection system trajectories performed during the lateral wall inspections of the shipping container cargo.

**Figure 11 sensors-23-00329-f011:**
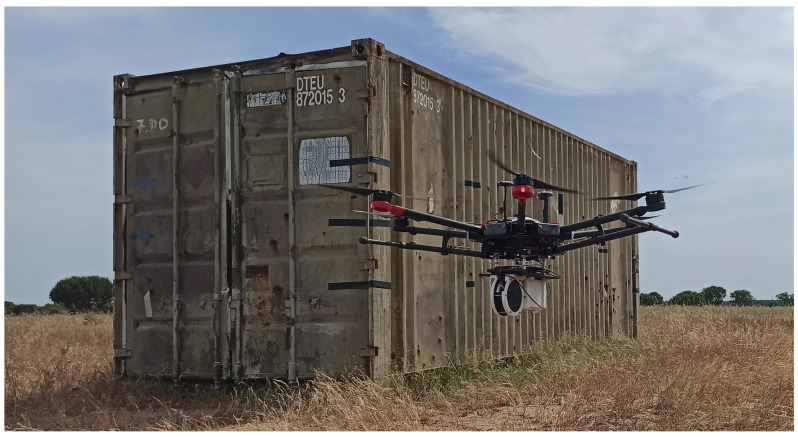
Multirotor with developed radiation detection system screening a shipping container cargo.

**Figure 12 sensors-23-00329-f012:**
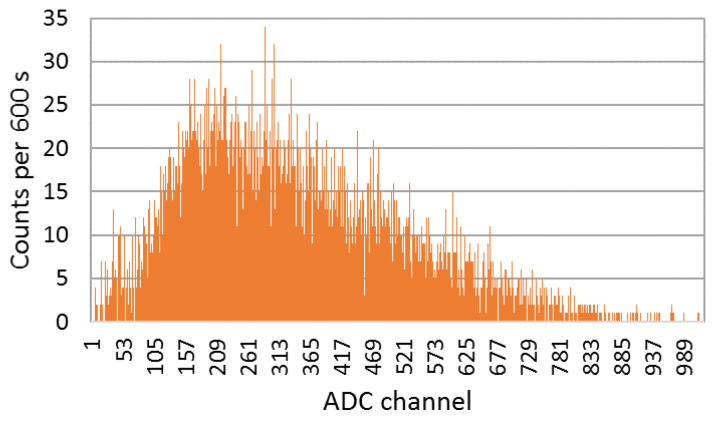
Spectrum from EJ-426HD detector considering a 33.3 MBq (0.9 mCi) Am-Be source at 2 cm distance.

**Figure 13 sensors-23-00329-f013:**
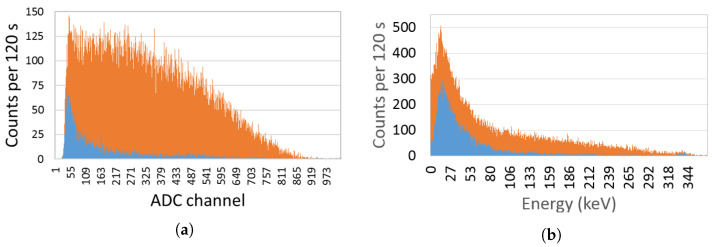
Beta spectra of a 3.3 kBq ^90^Sr source next to the detector window (orange color) and background (blue color) obtained with the: (**a**) EJ-200 scintillator (total counts: 61,561 ± 248; background counts: 5308 ± 73). (**b**) CsI(Tl) scintillator (total counts: 50,999 ± 226; background counts: 17,536 ± 132).

**Figure 14 sensors-23-00329-f014:**
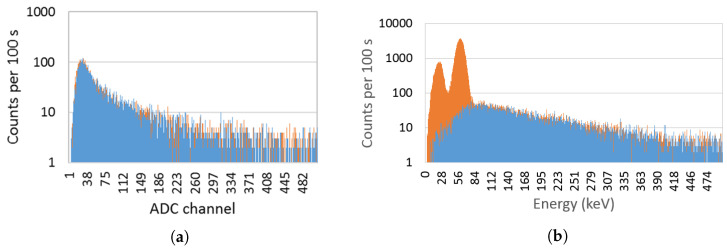
Gamma-ray spectra for a 22 kBq ^241^Am source next to the detector window (orange color) and background (blue color) obtained with the: (**a**) EJ-200 scintillator (total counts: 6269 ± 79; background counts: 6287 ± 79). (**b**) CsI(Tl) scintillator (total counts: 112,202 ± 335; background counts: 26,038 ± 161).

**Figure 15 sensors-23-00329-f015:**
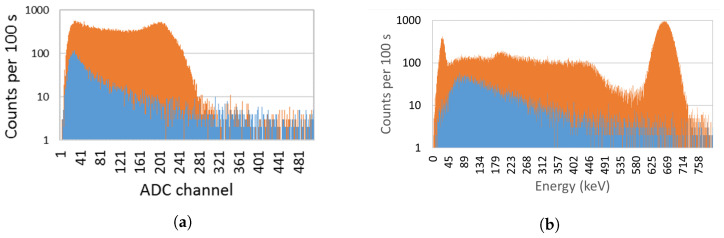
Gamma-ray spectra of a 8.5 kBq ^137^Cs source next to the detector window (orange color) and background (blue color) obtained with the: (**a**) EJ-200 scintillator (total counts: 181,780 ± 426; background counts: 12,538 ± 112). (**b**) CsI(Tl) scintillator (total counts: 310,116 ± 557; background counts: 26,038 ± 161).

**Figure 16 sensors-23-00329-f016:**
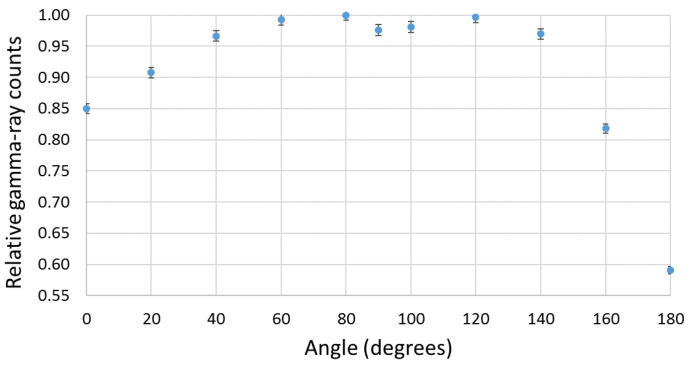
Gamma-ray detection efficiency varying the angle of a 0.11 MBq (2.86 µCi) ^137^Cs source relative to the EJ-200 scintillator (source–detector distance is kept constant).

**Figure 17 sensors-23-00329-f017:**
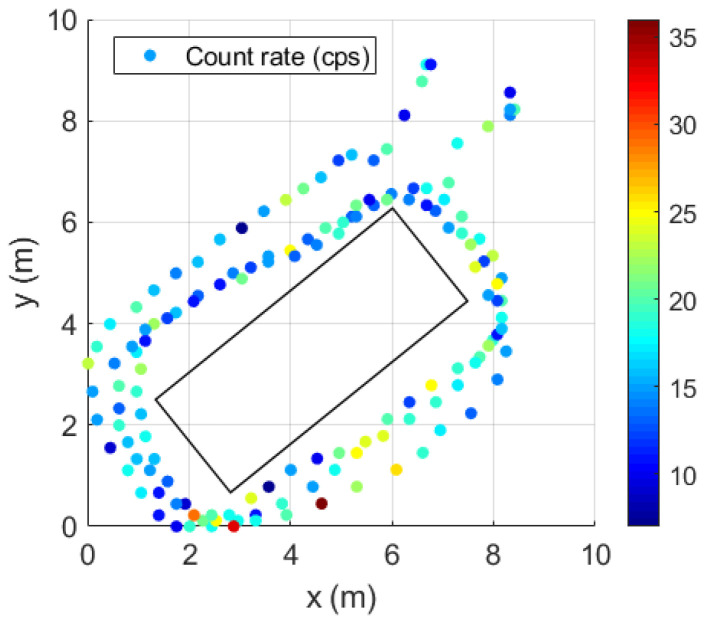
EJ-200 scintillator gamma-ray count rate (cps) of the background surrounding the shipping container.

**Figure 18 sensors-23-00329-f018:**
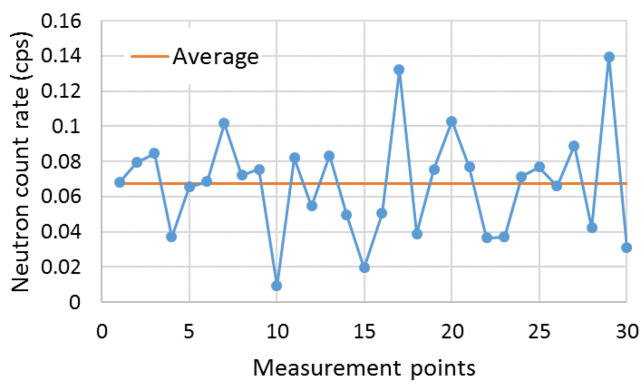
Neutron background count rate (cps) in the vicinity of the shipping container.

**Figure 19 sensors-23-00329-f019:**
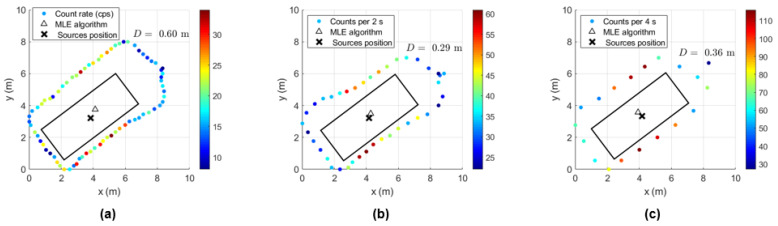
EJ-200 scintillator gamma-ray count rate (cps) measurements obtained using the handheld configuration of the detection system considering ^137^Cs sources of 4 MBq positioned at the center of the container. The distance “D” between the real sources position and the estimated position (MLE algorithm) is also given. Detection system trajectories were performed at half of the container height. Sampling times (dwell times) considered: (**a**) 1 s. (**b**) 2 s. (**c**) 4 s.

**Figure 20 sensors-23-00329-f020:**
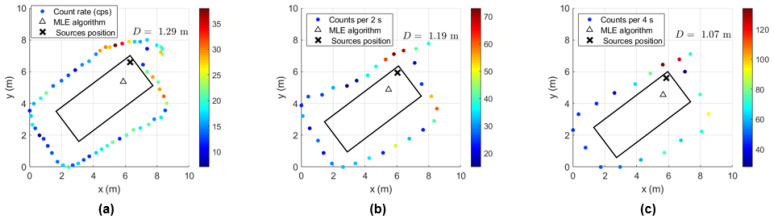
EJ-200 scintillator gamma-ray count rate (cps) measurements obtained using the handheld configuration of the detection system considering ^137^Cs sources of 4 MBq positioned at the bottom corner of the container. The distance “D” between the real sources position and the estimated position (MLE algorithm) is also given. Detection system trajectories were performed at half the container height. Sampling times (dwell times) considered: (**a**) 1 s. (**b**) 2 s. (**c**) 4 s.

**Figure 21 sensors-23-00329-f021:**
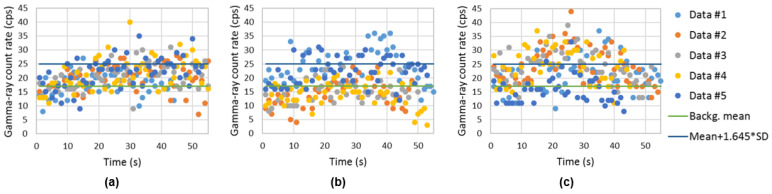
EJ-200 scintillator gamma-ray count rate (cps) measurements obtained using the detection system integrated in the drone considering ^137^Cs sources of 4 MBq positioned at the center of the container. The distance “D” between the real sources position and the estimated position (MLE algorithm) is also given. Drone trajectories were performed at: (**a**) one-third the container height. (**b**) half the container height. (**c**) two-thirds the container height.

**Figure 22 sensors-23-00329-f022:**
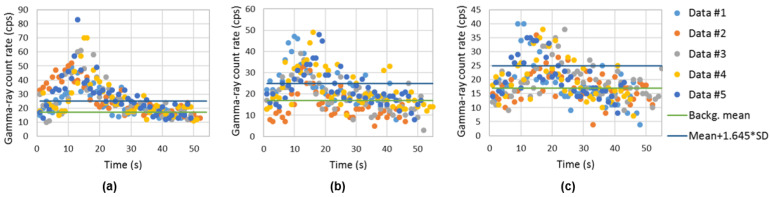
EJ-200 scintillator gamma-ray count rate (cps) measurements obtained using the detection system integrated in the drone considering ^137^Cs sources of 4 MBq positioned at the bottom corner of the container. The distance “D” between the real sources position and the estimated position (MLE algorithm) is also given. Drone trajectories were performed at: (**a**) one-third the container height. (**b**) half the container height. (**c**) two-thirds the container height.

**Figure 23 sensors-23-00329-f023:**
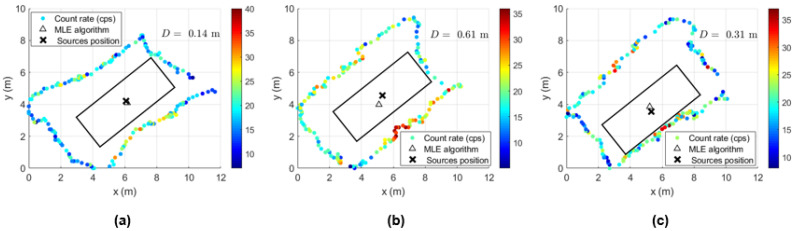
EJ-200 scintillator gamma-ray count rate (cps) measurements obtained using the detection system integrated in the drone considering ^137^Cs sources of 4 MBq positioned at the center of the container. The distance “D” between the real sources position and the estimated position (MLE algorithm) is also given. Drone trajectories were performed at: (**a**) one-third the container height. (**b**) half the container height. (**c**) two-thirds the container height.

**Figure 24 sensors-23-00329-f024:**
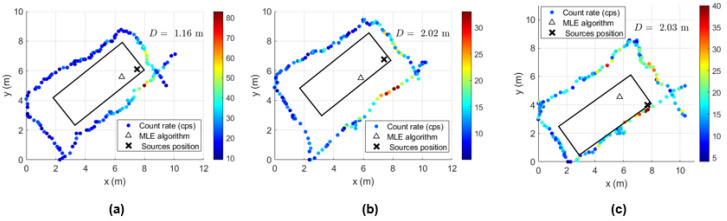
EJ-200 scintillator gamma-ray count rate (cps) measurements obtained using the detection system integrated in the drone considering ^137^Cs sources of 4 MBq positioned at the bottom corner of the container. The distance “D” between the real sources position and the estimated position (MLE algorithm) is also given. Drone trajectories were performed at: (**a**) one-third the container height. (**b**) half the container height. (**c**) two-thirds the container height.

**Figure 25 sensors-23-00329-f025:**
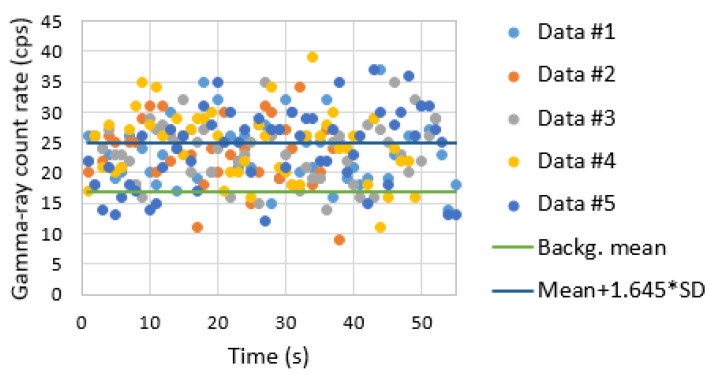
EJ-200 scintillator gamma-ray count rate (cps) measurements performed along the container length with detection system coupled with the drone considering the Troxler equipment oriented 90° (shielded 215 MBq ^137^Cs source—safe position) positioned at the center of the shipping container. Drone trajectory was performed at half the container height.

**Figure 26 sensors-23-00329-f026:**
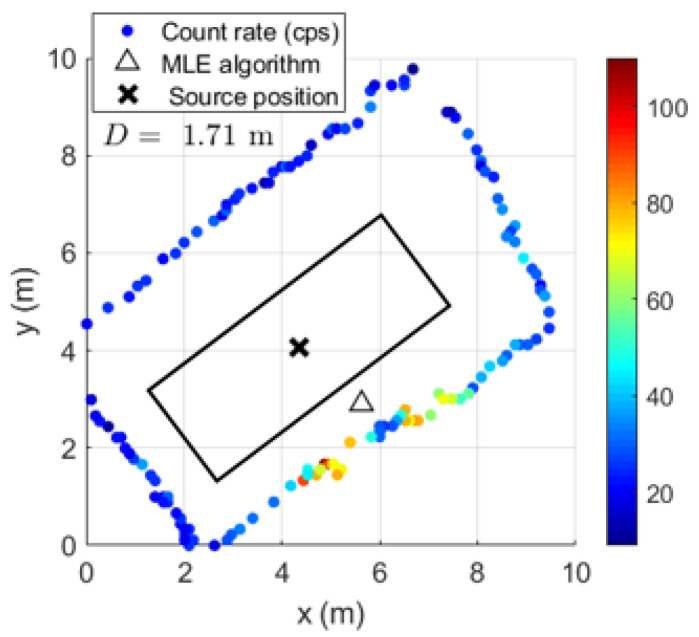
EJ-200 scintillator gamma-ray count rate (cps) measurements with detection system coupled with the drone considering Troxler equipment oriented 90° (shielded 215 MBq ^137^Cs source—safe position) in the center of the shipping container. The distance “D” between the real source position and the estimated position (MLE algorithm) is also given.

**Figure 27 sensors-23-00329-f027:**
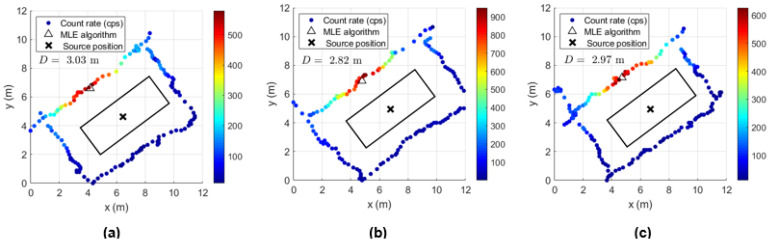
EJ-200 scintillator gamma-ray count rate (cps) measurements with detection system coupled with the drone considering the Troxler equipment oriented 90° (collimated 215 MBq ^137^Cs source exposed—first notch after safe position) in the center of the shipping container. Distance “D” between the sources position and MLE calculated position is also given. Drone trajectories were performed at: (**a**) one-third the container height. (**b**) half the container height. (**c**) two-thirds the container height.

**Figure 28 sensors-23-00329-f028:**
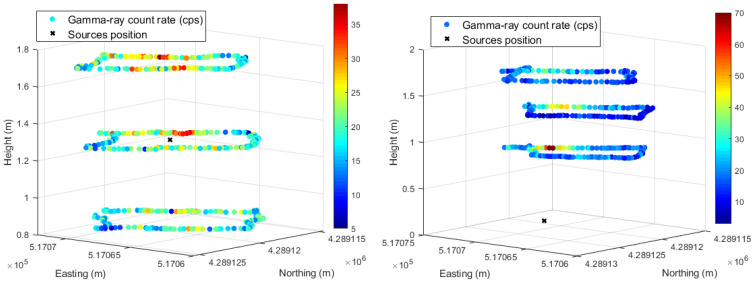
A 3D view of the EJ-200 scintillator gamma-ray count rate (cps) measurements considering the ^137^Cs sources of 4 MBq in the center (**left** figure) and at the bottom corner (**right** figure) of the shipping container. The true position of the sources is indicated by an “x” mark.

**Figure 29 sensors-23-00329-f029:**
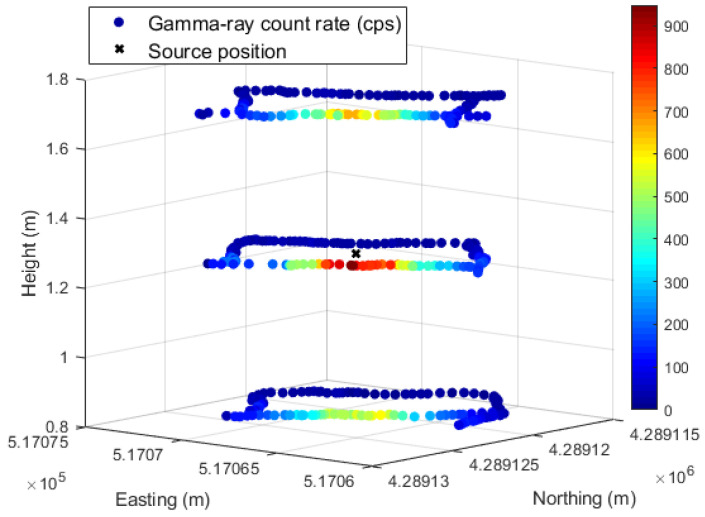
3D view of the EJ-200 scintillator gamma-ray count rate (cps) measurements considering the Troxler equipment (collimated 215 MBq ^137^Cs source—first notch after the safe position) in the center of the container. The true position of the Troxler is indicated by a “x”.

**Table 2 sensors-23-00329-t002:** Recent gamma-ray detection systems coupled to a multirotor.

Detector	Advantage	Limitation	Ref.
Geiger–Müller	Low cost; lightweight; dose rate measurements	No spectroscopy	[40,41,42,43]
CZT	Lightweight; high resolution	Small volume (∼1 cm^3^); expensive	[36,40,41,44,45,46,47,48,49,50]
NaI(Tl) with SiPMs	Compact; low power (<1 W) and compatible with strong magnetic fields	Medium resolution; Hygroscopic	[39]
CsI(Tl) with SiPMs	Lightweight; low power; higher light yield than NaI(Tl)	Medium resolution; slightly hygroscopic	[38,51]
BGO	High sensitivity; crystal volume available (103 cm^3^)	Poor energy resolution; heavy	[52]
CdTe	High energy resolution	Only for low-energy gamma rays	[53]
PIN Diode	Lightweight, low power consumption, and low cost	Small volume; susceptible to noise vibrations	[54]
CMOS (Timepix)	Lightweight, and low power consumption	Small volume	[55]

**Table 3 sensors-23-00329-t003:** Advantages and limitations of the use of plastic scintillators [15].

Advantage	Limitation
- Gross counting gamma rays (above 100 keV)	- Cannot be used for X-ray/gamma-ray spectroscopy
- Large size sheets and different shapes are available	- Light yield is one factor of 4 lower than that of NaI(Tl) scintillator
- Ruggedness and no regular maintenance	- Lower intrinsic efficiency than inorganic scintillators
- Good charged particle and neutron detectors	
- 500 times more efficient for detecting photons than a gas detector	
- Fast response	
- Low cost	
- Lightweight (lower density than inorganic scintillators)	

**Table 4 sensors-23-00329-t004:** Comparison of the MC simulation data (F8 tally) for the EJ-200 and CsI(Tl) detectors considering point sources of ^241^Am and ^137^Cs at different source–detector distances.

Source–Detector Distance	F8 Tally for EJ-200 (per Starting Particle)	F8 Tally for CsI(Tl) (per Starting Particle)	F8 Tally Ratio between EJ-200 and CsI(Tl)
^241^Am at 1 mm	(1.08 ± 0.01) × 10^−2^	(3.06 ± 0.01) × 10^−1^	0.035 ± 0.001
^241^Am at 1 m	(1.02 ± 0.03) × 10^−5^	(1.33 ± 0.01) × 10^−4^	0.077 ± 0.003
^241^Am at 5 m	(3.8 ± 0.2) × 10^−7^	(5.8 ± 0.1) × 10^−6^	0.066 ± 0.004
^137^ Cs at 1 mm	(1.99 ± 0.01) × 10^−1^	(3.23 ± 0.01) × 10^−1^	0.62 ± 0.01
^137^ Cs at 1 m	(2.03 ± 0.02) × 10^−4^	(1.28 ± 0.04) × 10^−4^	1.59 ± 0.06
^137^ Cs at 5 m	(8.4 ± 0.1) × 10^−6^	(5.7 ± 0.1) × 10^−6^	1.47 ± 0.04

**Table 5 sensors-23-00329-t005:** Neutron count rate and standard deviation obtained with the EJ-426HD detection system integrated in the drone considering Troxler equipment placed at the center of the shipping container.

Height (m)	Neutron Count Rate (cps) for Lateral Side Screening	Neutron Count Rate (cps) for a Complete Turn Screening
0.86	0.26 ± 0.05	0.24 ± 0.05
1.3	0.27 ± 0.06	0.26 ± 0.01
1.73	0.24 ± 0.08	0.22 ± 0.03

**Table 6 sensors-23-00329-t006:** Neutron count rate and standard deviation obtained with the EJ-426HD detection system integrated in the drone considering Troxler equipment placed at the bottom corner of the shipping container.

Height (m)	Neutron Count Rate (cps) for Lateral Side Screening	Neutron Count Rate (cps) for a Complete Turn Screening
0.86	0.45 ± 0.13	0.46 ± 0.09
1.3	0.34 ± 0.09	0.28 ± 0.03
1.73	0.37 ± 0.05	0.30 ± 0.06

**Table 7 sensors-23-00329-t007:** Average and SD of the distance (D) between the MLE estimated position and the true source position using the gamma-ray counts of EJ-200 scintillator considering ^137^Cs sources of 4 MBq placed at the center and at the bottom corner of the shipping container.

Sampling Time (s)	D (m)—Sources Centered	D (m)—Sources at the Corner
1	0.84 ± 0.31	1.3 ± 0.3
2	0.44 ± 0.22	1.1 ± 0.4
4	0.47 ± 0.07	1.1 ± 0.1

**Table 8 sensors-23-00329-t008:** Average and SD of the distance (D) between the estimated position of the source (using MLE algorithm) and the true source position using the gamma-ray counts of EJ-200 detector considering ^137^Cs sources of 4 MBq placed at the center and bottom corner of the shipping container.

Height (m)	D (m)—Sources Centered	D (m)—Sources at the Corner
0.86	0.76 ± 0.39	1.56 ± 0.28
1.3	0.90 ± 0.46	2.22 ± 0.99
1.73	0.41 ± 0.13	2.54 ± 0.36

**Table 9 sensors-23-00329-t009:** Average and SD of the distance (D) between the estimated position of the source (using MLE algorithm) and the true source position using the EJ-200 scintillator gamma-ray count rate measurements considering the Troxler equipment oriented 90° (collimated 215 MBq ^137^Cs source exposed—first notch after safe position) in the center of the shipping container.

Height (m)	D (m)—Source Centered
0.86	2.96 ± 0.37
1.3	2.93 ± 0.20
1.73	2.77 ± 0.19

## Data Availability

Not applicable.

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
