# Peer review of "Neutron and Gamma-Ray Detection System Coupled to a Multirotor for Screening of Shipping Container Cargo"

_sensors, 2022, doi:10.3390/s23010329_

Round 1

Reviewer 1 Report

The paper presents an interesting application. However, the following important improvements should be applied:

1)     The length of the manuscript appears excessive (34 pages!). I suggest to shorten the presentation of experimental results and put some plots in the supplementary material.

2)     Among 70 references, only a few are to papers in Sensors journal. I recommend to add the citation to a recent paper https://doi.org/10.3390/s22041412 presenting a compact CsI gamma ray detector embarked on a drone.

3)     More details about the readout of SiPM should be added. What type of circuit is used in the Topaz system? What SiPM are used? Are they integrated with scintillators by the EJ company?

4)     The connection between detectors and electronics is not clear (neither the description in the text lines 278..., nor Fig. 1 where the detectors are in series). How can the neutron detector be connected to a GPIO digital pin instead of an analog input? Please fix all this section.

5)     How was the size and arrangement of detector chosen?

6)     For Energy axis of spectra please use Energy (calibrated in keV) rather than ADC channels.

7)     What about the power consumption of the system? What about the battery and its lifetime? What about the flight time of the drone when lifting this system?

Author Response

The authors thank the reviewer for her/his insightful comments and suggestions that contribute to increase the overall quality of the manuscript.

The paper presents an interesting application. However, the following important improvements should be applied:

  1. The length of the manuscript appears excessive (34 pages!). I suggest to shorten the presentation of experimental results and put some plots in the supplementary material.

Response 1: The authors thank the reviewer for this comment. The size of this article is related to the wide variety of scientific and technological topics addressed in the manuscript  (related to gamma and neutron detection systems, encompassing  inter alia measurements, Monte Carlo simulations, instrumentation, electronics and IT topics). We fear that placing the experimental results as supplementary material might arguably break the reader´s sequence and could negatively impact the overall presentation and discussion of the broad spectrum of topics discussed. According to https://www.mdpi.com/journal/sensors/instructions “Sensors has no restrictions on the length of manuscripts, provided that the text is concise and comprehensive”. Also says “The recommended length of an Article is more than 16 journal pages”.

  1. Among 70 references, only a few are to papers in Sensors journal. I recommend to add the citation to a recent paper https://doi.org/10.3390/s22041412 presenting a compact CsI gamma ray detector embarked on a drone.

Response 2: Thank you very much. 

Added the reference below to the bibliography:

Carminati, M.; Di Vita, D.; Morandi, G.; D’Adda, I.; Fiorini, C. Handheld Magnetic-Compliant Gamma-Ray Spectrometer for Environmental Monitoring and Scrap Metal Screening. Sensors 2022, 22, 1412.

The reference is cited in the following text (added to l. 185): “The use of a SiPM based scintillator was also demonstrated for the detection of radioactive sources in scrap metal (waste and recycle material monitoring) when strong magnetic fields (0.1 T) are present.” and in table 2.

  1. More details about the readout of SiPM should be added. What type of circuit is used in the Topaz system? What SiPM are used? Are they integrated with scintillators by the EJ company?

Response 3: Thank you for this very relevant comment. 

The following texts were added:

Relative to EJ-200 scintillator: “(...) four 12 x 12 mm2 SiPMs (arrays J-60035-4P-PCB).“ (l. 264)

Relative to CsI(Tl) scintillator: “(...) two 12x12 mm2 SiPMs (arrays J-60035-4P-PCB) and an aluminum housing.” (l. 270)

Relative to EJ-426HD neutron detector: “(...) three 6x6 mm2 SiPMs (KETEK PM6660).” (l. 274)

The following text “All detectors were produced by Scionix and the specifications are resumed as follows:”  (l. 260) was replaced by:  “ All detectors were manufactured by Scionix (including the SiPMs integration on the scintillators) and the specifications are resumed as follows:”

The following text was added: “TOPAZ-SiPM multichannel analyzer (MCA), developed by BrightSpec [61], with a power consumption of approximately 1.1 W has three input connectors: i) a Lemo connector (type ERN.03.302.CLL) to read the detector analog signals and to provide the necessary power to the SiPMs integrated on the scintillators (5V, 20 mA); ii) a Lemo connector (type ERN.00.250.CTL) for a programmable general purpose input/output (GPIO) signals (can be used as an external counter input); and iii) a USB type mini B for data output, device power supply and control using for example a Raspberry Pi model 3B. TOPAZ-SiPM MCA combines in a small and lightweight box (70 mm x 45 mm x 26 mm, 70 g) the following features: analog to digital converter (ADC) with a spectral memory size up to 4096 channels, analog signal amplification (up to 16), a traditional trapezoidal shaper for digital pulse processing, a digital baseline restorer and a pile-up rejector, and a 5V low-ripple (low-noise) power supply for the SiPMs preamplifiers.”

  1. The connection between detectors and electronics is not clear (neither the description in the text lines 278..., nor Fig. 1 where the detectors are in series). How can the neutron detector be connected to a GPIO digital pin instead of an analog input? Please fix all this section.

Response 4: Thank you. In order to clarify the connection scheme and the GPIO digital pin, the paragraph between lines 279 and 283 was changed to:

 “Since only one TOPAZ-SiPM MCA was available, to simultaneously read the gamma-ray/beta and neutron detection system signals it was necessary to connect the EJ-200 scintillator to the analog input of TOPAZ-SiPM MCA (Lemo connector type ERN.03.302.CLL) and the EJ-426HD neutron detector (TTL output) to the GPIO input of TOPAZ-SiPM MCA (Lemo connector ERN.00.250.CTL). When using the TTL output of the EJ-426HD neutron detector its analog output (LEMO connection) is only used for power supply purposes (connected to a +5 V power plug available in the EJ-200 housing). In order to obtain the energy spectrum of the EJ-426HD neutron detector it is also possible to connect its analog output into TOPAZ-SiPM MCA (only used on laboratory tests), however in this case EJ-200 cannot be connected to TOPAZ-SiPM MCA (analog connector already in use).”

The following text was added: “(...) each TTL pulse corresponds to a neutron count (detector internally adjusted above noise at 40 degrees C)” (l. 271)

Figure 1 was changed for better understanding the power connection between the EJ-426HD and the EJ-200 detectors (are in parallel in fact).

  1. How was the size and arrangement of detector chosen?

Response 5: Thank you. The following text was added (l. 275): “The detectors size and arrangement were chosen according to three factors: maximize the detection efficiency, not exceed the platform´s maximum take-off weight and fit on the carbon fiber sandwich sheet developed to carry the gamma and neutron detection system side-by-side.”

  1. For Energy axis of spectra please use Energy (calibrated in keV) rather than ADC channels.

Response 6: We fully recognize the pertinence of the reviewer's comment. The x axis of the CsI(Tl) spectra were changed to “Energy (keV)” - Figures 13b, 14b and 15b . For the x axis of the EJ-200 scintillator it was chosen to keep “ADC channels” because this detector does not have photopeaks to do a proper calibration.

  1. What about the power consumption of the system? What about the battery and its lifetime? What about the flight time of the drone when lifting this system?

Response 7: Thank you. The following texts were added: 

“The radiation detection system and associated electronics have a power consumption of approximately 2.75 W (550 mA current, 5 V). Using a power bank of 10 Ah a battery life of up to 18 h was obtained.” (l. 287)

“For the considered radiation detection system (payload), a flight time of 17-22 minutes was achieved (depending on the path performed by the drone and battery pack used).”  (l. 476)

Revisions were also marked up using the “Track Changes” function. 

Reviewer 2 Report

The paper is well written and contains detailed experimental procedure and the subsequent results.

I believe the paper may be accepted as is.

Author Response

The paper is well written and contains detailed experimental procedure and the subsequent results. I believe the paper may be accepted as is.

Response: Authors thank the reviewer for her/his comments and paper acceptance.

Revisions were also marked by using "Track changes" function.

Reviewer 3 Report

  1. Page 7 line 262-274, could the author add information of the model, area and working conditions (bias, gain, and QE) of SiPM.
  2. Fig 12-14, more information is needed to understand the meaning of x-axis. 
  3. In section 3.4.3, the gamma ray detection measurement assumes a source placed in the container without other material. Could the author comment on the accuracy in the case of more material around the radiation source since the gamma ray would be attenuated by added material.
  4. Fig 21 and 22, what’s the meaning of x axis and what the meaning of different colors in the plots?
  5. Fig 22, what caused the peak structure around 15 seconds? Does that correspond to the time when the drone is close to the source?
  6.  

Author Response

Authors thank the reviewer for her/his insightful comments and suggestions that contribute to increase the overall quality of the manuscript.

  1. Page 7 line 262-274, could the author add information of the model, area and working conditions (bias, gain, and QE) of SiPM.

Response 1: Thank you. The following texts were added:

Relative to EJ-200 scintillator: “(...) four 12 x 12 mm2 SiPMs (arrays J-60035-4P-PCB).“ (l. 264)

Relative to CsI(Tl) scintillator: “(...) two 12x12 mm2 SiPMs (arrays J-60035-4P-PCB) and an aluminum housing.” (l. 270)

Relative to EJ-426HD neutron detector: “(...) three 6x6 mm2 SiPMs (KETEK PM6660).” (l. 274)

TOPAZ-SiPM MCA provides the necessary working conditions for the SiPMs (e.g. voltage and current). Some parameters (like the gain and the QE) are specific for a given SiPM model, therefore can be consulted in the datasheet. The only parameters that can be changed are the ones referred in subsection 2.2 (step 3 – Hardware initialization) and are essentially related to TOPAZ-SiPM MCA parameters (signal processing optimization).  

  1. Fig 12-14, more information is needed to understand the meaning of x-axis. 

Response 2: Thank you. The following information was added to the x-axis: “ADC channel”.

  1. In section 3.4.3, the gamma ray detection measurement assumes a source placed in the container without other material. Could the author comment on the accuracy in the case of more material around the radiation source since the gamma ray would be attenuated by added material.

Response 3: Thank you for this very relevant comment. The following text was added: 

“Moreover, if container cargo material was considered around the radiation source more gamma-ray and neutron attenuation would take place (depending on the material density and atomic number) reducing the counting rate measured outside and consequently the MDA and the radiation source localization accuracy.” (l. 620)

  1. Fig 21 and 22, what’s the meaning of x axis and what the meaning of different colors in the plots?

Response 4: The different colors correspond to the five lateral screenings performed (five for each graph). A legend was added to the graphs of figure 21 and 22.

The following text was also added: “The x axis is related to the time elapsed since the beginning of the container screening. Different colors were used to distinguish the five screenings performed.” (l. 586)

  1. Fig 22, what caused the peak structure around 15 seconds? Does that correspond to the time when the drone is close to the source?

Response 5: Thank you. The following text was added: “The peak registered on the graphs of Figure 22 at approximately 15 s (in particular the screening performed at one third the container height) corresponds to the time the drone is closer to the source (pass through the source).” (l. 598)

Revisions were also marked by using the "Track changes" function.

Round 2

Reviewer 1 Report

The revision has addressed my concerns.